# Investigation of the Geometrical Deterioration of Paved Superstructure Tramway Tracks in Budapest (Hungary)

Vivien Jóvér [1], Zoltán Major [1,*], Attila Németh [1], Dmytro Kurhan [2], Mykola Sysyn [3] and Szabolcs Fischer [1,*]

1 Central Campus Győr, Széchenyi István University, H-9026 Győr, Hungary; jover.vivien@sze.hu (V.J.); nemeth.attila@sze.hu (A.N.)
2 Department of Transport Infrastructure, Ukrainian State University of Science and Technologies, UA-49010 Dnipro, Ukraine; d.m.kurhan@ust.edu.ua
3 Department of Planning and Design of Railway Infrastructure, Institute of Railway Systems and Public Transport, Technical University of Dresden, 01069 Dresden, Germany; mykola.sysyn@tu-dresden.de
* Correspondence: majorz@sze.hu (Z.M.); fischersz@sze.hu (S.F.); Tel.: +36-(96)-613-544 (S.F.)

**Abstract:** In the 21st century, one of the key requirements is to develop and maintain our infrastructure facilities most efficiently using the available resources. Tramways are of significant national economic importance and represent an important national asset. There are currently seven different types of superstructure systems in Hungary, based on the national regulations and the related requirements currently in force. This paper compares the paved tramway superstructure systems in the context of track geometry, through-rolled axle tons of track, and the age of track sections. Paved tracks have many benefits, but the main ones are easier maintenance and road traffic use. Elastically supported continuous rail bedding (ESCRB; in Hungary, this is known as "RAFS") and "large" slab superstructure systems are used to create paved superstructure systems. Road crossings use the latter systems, while heavily loaded lines use several ESCRB systems. This article examines the geometrical changes in several ESCRB superstructure systems. A TrackScan 4.01 instrument was used to take measurements in June and September 2021 and in April 2022, September 2022, and May 2023. Track gauge, alignment, and longitudinal level are examined. Regardless of the ESCRB superstructure system or age, a medium-loaded line's track gauge trendline increases, which means that the track gauge is widening and, regardless of traffic load or age, the average longitudinal level is constantly increasing from year to year. When it is a medium-loaded line, the average value of alignment increases slightly, and the trendline is almost straight, but it decreases when it is an extremely heavily loaded line. The authors will analyze how the reference track section will change in the future. Based on the results, it is important to assess how subsequent measurements affect the trend lines. Because the data evaluations show similar results, comparing open tramway tracks to paved ones is crucial.

**Keywords:** tramway; deterioration; paved track; ESCRB; RAFS; TrackScan 4.01; geometrical analysis; traffic load



## 1. Introduction

The authors should clarify the specific layout of the Introduction. This comprises a three-part introduction: Section 1.1 discusses transportation and fixed-rail transportation. Section 1.2 examines the international literature on tramways. Section 1.3 discusses the study's innovation, significance, and setup. Last, but not least, the authors must emphasize that the current paper is the continuation and supplementary article of a previously published paper [1].

### 1.1. General Introduction

Transport is often divided into three categories: (i) land transport, (ii) air transport (including space travel), and (iii) water transport (shipping or navigation). Each location

may be divided into passenger and freight transportation. For example, individual travel and public transport are theoretically possible in every area, but rail and space flight are technologically and legally restricted. Hence, even if the issue of freight transport is solved and is successful in all locations, space travel, wherein freight transport is still evolving, may be the most exciting issue to research. (The authors must highlight that nowadays, the transportation and travel needs of people with disabilities are one of the most important areas of transportation research [2,3]).

Transport sciences include transport engineering, with logistics–transport–packaging sub-disciplines, civil engineering, mining and metallurgical engineering, urban, municipal, and architectural engineering, mechanical engineering, electrical engineering, and vehicle engineering. Some of the above disciplines are worth mentioning as these sectors often overlap [4–8].

Fixed-rail transit is essential to land mobility. It includes railroads, tramways, detached railways (undergrounds, elevated railways, local-interest railways, etc.), rack railways, cableways, and funicular railways. Rail travel gained popularity internationally in the early 1800s. Railways carry vast amounts of freight, bulk commodities, and passengers across both short and great distances. European statistics [9,10] show that in 2018, the EU-28 (the 28 member nations of the European Union) had 407.2 billion passenger-kilometers and 423.3 billion ton-kilometers of rail freight (a total values were 5915.9 billion and 2371.2 billion, respectively).

The current article discusses civil engineering, which deals with tramway tracks (it should be mentioned that tramways use mainly electric hauling models with catenary systems or third-rail sunken systems). Public railways are explained in Section 1.2.

*1.2. Literature Review*

Nowadays, the most widely used superstructure system in the world is the ballasted track, not only for railways but also for tramway tracks. In the capital of Hungary, Budapest, traveling by tramway is one of the most critical public transport options [11]. The tramway network covers almost the entire city: tracks of nearly 300 km long, six types of vehicles, and seven types of superstructure systems are used. Tramway transits differ from public (large-scale) railroads due to their different speeds and axle weights [11]. (Note: The authors felt that a comparison between the high-speed railway and the heavy haul railway was necessary in this article because, although the tramways differ significantly in both axle load and design and in clearance speed, they can be compared in terms of their character and aging process. Of course, the authors are aware that tramways may present significantly different defects but, in many respects, the deterioration process is similar. However, the international literature on tramway track geometry deterioration is relatively scarce. Thus, they argue that it was essential to include these two areas in the literature search).

Due to the growth of the city's population and the appearance of more and more cars, motorcycles, and bicycles, the railway track had to be treated as part of the city. Using two side-by-side open superstructure system tracks, which are closed off from the road, was no longer successful in the city center.

Recognizing the problem, the Budapest Transport Privately Held Corporation (BKV Ltd., Budapest, Hungary; in Hungarian, it is BKV Zrt., Budapest, Hungary), the company responsible for operating and maintaining the railway tracks, started using paved superstructure systems at the beginning of the 2000s. Among their many advantages, one of the most important is that road traffic can also use the same route. An additional advantage is that the ballasted bed does not get dirty and is easy to maintain, but the disadvantage is that it is difficult to estimate the condition of the track. Nowadays, 55% of the tramway network comprises a paved superstructure system in Budapest.

There are two types of paved superstructure systems: an elastically supported continuous rail bedding system (ESCRB; in Hungary, the so-called "RAFS" abbreviation is applied) and the "large" slab superstructure system. The latter is used in road crossings, and several ESCRB systems are used for the most heavily loaded lines.

Related to the above ideas, the authors have prepared a detailed international literature review on tramways, presented in the following paragraphs. At the beginning of each more extended block, the main sub-topic to which the discussion relates is indicated as a subsubsection (i.e., Sections 1.2.1–1.2.4).

### 1.2.1. General Ideas about Tramways and Public Transportation Systems in Cities

Europe's largest cities need an excellent public transportation system [12,13]. Public transportation networks make communities more livable and lucrative while decreasing pollution [14]. Energy efficiency and electric or electric–hybrid fixed-rail vehicle services are vital due to rising worldwide electric energy consumption [15,16]. Most cities rely on guided (fixed-rail) land transit [11,17,18]. Infrastructure and public transit users must cooperate as metropolitan populations grow. Consequently, tramway safety technologies must detect, identify, and monitor people, cars, and bikes [19,20].

A substantial increase in power and fuel costs [21] (e.g., gas, oil, kerosene, etc.) would challenge both countries and the public, as well as privately owned public transport corporations, in 2021–2023 (the same can, of course, be said for private transport, with the rising prices of petrol, diesel, electricity, natural gas, etc.). Of course, many factors exist: the ongoing war in Eastern Europe [22], the COVID-19 epidemic [23], and the global financial market's well-documented boom and bust are other causes. Nevertheless, the most likely solution combines the components above in varying degrees.

In an urban infrastructure, the inhabitants must tolerate train noise to live safely and comfortably [17,18,24–26]; e.g., rail dampers are cheap and quiet, but their performance characteristics are unclear [27,28].

### 1.2.2. Track Gauge as One of the Most Relevant Parameters of Fixed-Rail Systems

A study on Zagreb's urban traffic planning recommends an LRT (light rail transit) system to fulfill current and future needs and improve the service [29]. The analytic hierarchy process (AHP) is used to select the best track gauge in the aforementioned study, which advocates an underground and above-ground LRT system. The authors discuss whether to adopt the 1000 mm gauge used by the current tram network, which has lower construction costs, or a 1435 mm gauge, as in most other cities, which incurs lower long-term maintenance expenses. An AHP analysis with specific criteria and sub-criteria for future study is then suggested. They report that the 1435 mm gauge (53.9%) outperforms that of the 1000 mm gauge (46.1%). The article suggests an AHP analysis technique, along with specific criteria and sub-criteria for further investigation [29].

Puffert [30] used a spatial network model utilizing agent choices to study regional standard railway track gauge history. The UK, continental Europe, North America, and Australia standardized their systems based on contingent events and positive feedback. Regional gauges were mostly arbitrary, but connecting lines utilized the same gauges for compatibility. Early diversity and network integration benefits, set against conversion costs, shaped these places' diversity resolution. As with other gauges, the gauge used on 60% of the world's railways arose through incidental occurrences, and even historical mishaps, and was then reinforced by positive feedback rather than by underlying reasoning, systematic optimization, or market testing [30].

Almech et al. [31] suggested merging the three subnetworks into two and linking them with gauge adjustments to discover the shortest path in the Spanish railway network. This unique technique calculates the shortest rolling stock path using the dual gauge subnetwork, Iberian, and standard gauge networks. Dual gauge railway networks simplify finding the shortest route and time computation; thus, any classic shortest-path algorithm can solve the problem [31].

### 1.2.3. Geometrical Measuring Possibilities as well as the Monitoring of Tramways

Tang et al. [32] proposed a laser triangulation-based gauge detection system including a photoelectric encoder, 2D laser scanner, Ethernet switch, and data processing system.

Detection precision exceeds the system accuracy, which is ± 0.6 mm. Gauge point extraction accurately computes the track gauge readings, according to DIN EN 14811: 2010 [33]. The gauge identification system accurately identifies grooved rail profiles [32].

Madejski [34] illustrates why track and turnout geometry information are important and how to use synthetic quality parameters to evaluate them. Catenary data may be used to improve infrastructure information databases. Track and switch geometry must be measured over time. The diagnostic module flags the track's "weak regions" for inspection. Many tram, metro, and light rail operators and maintenance businesses employ synthetic assessments with speed-based tolerance limits. Synthetic track quality coefficients help make longer track section overhaul decisions economically [34].

Modern tramways are Kraków's biggest transit enhancement [35]. Laser scanning is used to speed up the maintenance of the transportation infrastructure. Data quality was verified by post-processing the Kraków Rapid Tram Tunnel measurements. Scan data management is performed online; laser scanning, surveying, and analytics monitor the urban tramway networks today and, possibly, in the future. Laser scanning helps in both industrial and communication networks. To analyze structural gauges, tramway rails, and the infrastructure, surveys and diagnostics require tramway track location data. Designing and making the relevant laser scanning measurements requires point cloud resolution, station localization and distance, and the objective verification of laser scanning data accuracy and quality versus conventional equipment. Field surveying control and tram track maintenance have been improved. Reference sites are surveyed and diagnosed using magnetic reflectors or scanning targets [35].

The ZET Municipality maintains 116 km of tramway tracks in highly populated Zagreb areas [26]. The Zagreb Faculty of Civil Engineering built a floating slab track to dampen the vibrations since rail running surface flaws and discontinuities enhance vibrations. Worker vibration analysis is required by the 2002/44/EC standard [36]. Periodic vibration readings aid in both decision-making and maintenance [26].

A pseudo-excitation (PEM) study examines tram–track interaction on a curved track due to polygonal wheels and track irregularity [37]. Field testing confirmed the presence of dynamic behavior with rail vibration, deformation, and the wheel wear profile. PEM enhanced wheelset flexibility, dynamic wheel-rail contact force, creepage, rail, car-body, and axle-box vibration modes. A dynamic tram vehicle–curved slab track model examined tram-track random vibrations on a curve. The simulation showed that wheelset flexibility, first torsional, and second bending modes affect rail, car-body, and axle-box vibration and increase wheel–rail contact force and creepage. The tram suspension systems' inherent frequency of 10 Hz, vehicle transitions from straight to curved tracks (t = 1.3), and polygonal tire wear and track irregularity frequencies of 114 Hz generate car–body vertical and lateral vibrations [37].

Reducing the analytical segment length affected tram network condition assessment in Osijek, Croatia [38]. Track geometry quality can be assessed by comparing the measured parameters to the tolerances or by calculating the TQI (track quality index) for a track segment of a certain length. The five narrow-gauge tram-track geometry parameters from the Osijek tram network in Croatia were much lower than the allowable tolerances for the observed values under loaded track conditions. To address these discrepancies, the synthetic track quality indices, W (based on the analysis of the total effect of all five measured values of geometry parameters along each segment of the track, considering track gauge, superelevation, twist, and horizontal and vertical irregularities) and J (a quantitative track evaluation), were used to measure track quality. The J index used weighted standard deviations of the track geometry parameters, while the W index used geometric values beyond the authorized ranges. Track segmentation is used to evaluate tram track geometry and plan track repair or rebuilding. The W index is less affected by segment length than the J index; however, increased analytical segment length increases the W value of the segments. Segmentation must match up with tram track maintenance and repair. Segments are used to model the track conditions to reduce predicted tram track maintenance [38].

### 1.2.4. Deterioration of Tramways

Paved tramway track degradation may be structural or geometric. Paved railways first degrade in terms of the track gauge, then in alignment and longitudinal level, then the rails, fasteners, and so on [39–41].

Sudden acceleration shifts in speed can reveal the tramways' rail geometry. The acceleration data also assess passenger comfort and rail conditions. Thus, certain methodologies, tools, and acceleration data will aid in parameter monitoring [42].

Superstructure systems need early track damage detection systems, as well as regular repair and maintenance [43]. Recent studies have defined the railway track lifespan. Technical, economic, service, and moral lifetimes are commonly used. Morocco and other nations are studying tramway passengers' attitudes [44].

Paved ballasted railways typically last 30–50 years. However, different structural elements will result in different lifespans, which may vary [45].

Tramway rails have been the subject of fewer longevity studies than major railroad tracks. While fixing longevity issues, the track gauge and its variations are examined. Track gauge variance, traffic information, and structural factors were used to predict Melbourne tramway track deterioration [46]. Researchers used machine learning to predict the track degradation index (TDI), based on the data and subsequent analyses. The scheduling of repairs and procedures requires the calculation of future TDI values [47].

Experience dictates the choice of Budapest tramway tracks for study, but their geometrical degradation, durability, and lifecycle cost are unknown. Technically and economically, freshly erected superstructures' geometric qualities, potential load, and future maintenance and operation chores are important. Preventing projected cost increases is important, given current economic concerns. Risk management must be a project priority. Transport infrastructure growth boosts economic and social development [48].

Graz University of Technology researchers study train track endurance [49–51]. In Professor Peter Veit's 1997–2003 Strategie Fahrweg study project, the lifetime costs of railway lines and segments of railway tracks under various circumstances, geometries, and loads are computed for the entire OBB (Austrian State Railways) network [49–52]. Good and medium-to-poor substructures have lifetime costs of 1:3 to 1:9 (substructures need drainage, set load capacity, etc.). Additionally, when maintaining a railway turnout with a 500-m radius and a length of 42 m, the costs equal to a 450-m length straight track's maintenance costs. Traffic disturbance (especially single-track traffic) and depreciation charges are half the cost or more of a railway track segment. Curvature matters. A short radius curve's lifetime cost may be double that of a straight route. Lifetime engineering is an important field of engineering. Gáspár and co-authors have published many articles on this topic [45,53,54]. Sarja [55] introduced the concept of lifetime engineering.

ANN (artificial neural network) and SVM (support vector machine) models can be used to predict Melbourne tram gauge degradation [56]. Filtering out-of-range and incomplete data improve the models' accuracy. Two machine learning algorithms predict track gauge fluctuation, affecting riding comfort and derailment [56].

### 1.3. The Novelty and Structural Setup of the Current Paper

This article is a continuation of the authors' previous articles [1,57–59]. The focus of this analysis is on paved tramway tracks. In January of this year, an article on open tramway tracks was prepared using nearly identical principles [1]. Paving in historic city center areas, grade crossings, and sections that can be used by road vehicles constitutes a significant portion of Budapest's tramways; therefore, this research topic must be addressed. This paper analyzes the deterioration and temporal variation of the track geometry parameters based on our own Trackscan instrumentation measurements, which were scheduled to be conducted four times in 2021 and 2022. The future objective of this research series is to determine with relative precision how each superstructure's structural design degrades over time. As mentioned previously, Budapest has seven distinct types of superstructures, which can be grouped into two main categories: open and paved. According to the authors,

such a comprehensive study has never been conducted; therefore, this mathematically based statistical analysis could significantly contribute to the international literature.

Last but not least, the authors must mention (see Section 1) that the current paper is a continuation and supplementary article of a previously published paper [1].

Section 2 presents the materials and methods used in this study, Section 3 presents the results, Section 4 offers a discussion, and finally, Section 5 details the authors' conclusions.

## 2. Materials and Methods

### 2.1. Examined Paved Reference Sections

The examined reference track sections can be grouped into three different superstructure systems; these are as follows:

- ESCRB I track system: Section #1, Section #2, and Section #3;
- ESCRB II track system: Section #4, Section #5, Section #6, and Section #7;
- ESCRB III track system: Section #8, Section #9, Section #10, and Section #11.

The similarity between these superstructure systems is that they all have an elastically supported continuous rail bedding system and are paved tracks.

The ESCRB I superstructure system (Figure 1) has a reinforced concrete slab of 28 cm in depth. The grooved rails or flat-bottom rails (i.e., the so-called Vignole rail profiles) are built with rubber profiles and are stabilized by fastenings with limited clamping force. The fastenings are constructed as follows: the dowel screws, which are equipped with nuts and bolts, must be pressed together with the tie plates onto the rail foot rubber band. Around the tie plates and in the sections between them, formwork must be applied, and then the pour should be carried out—usually with Gantrex material. This fastening mimics the so-called Gantry rail fastening used in Hungary. Between the rails, there are steel gauge holder rods; these are present to ensure an accurate (adequate) track gauge.

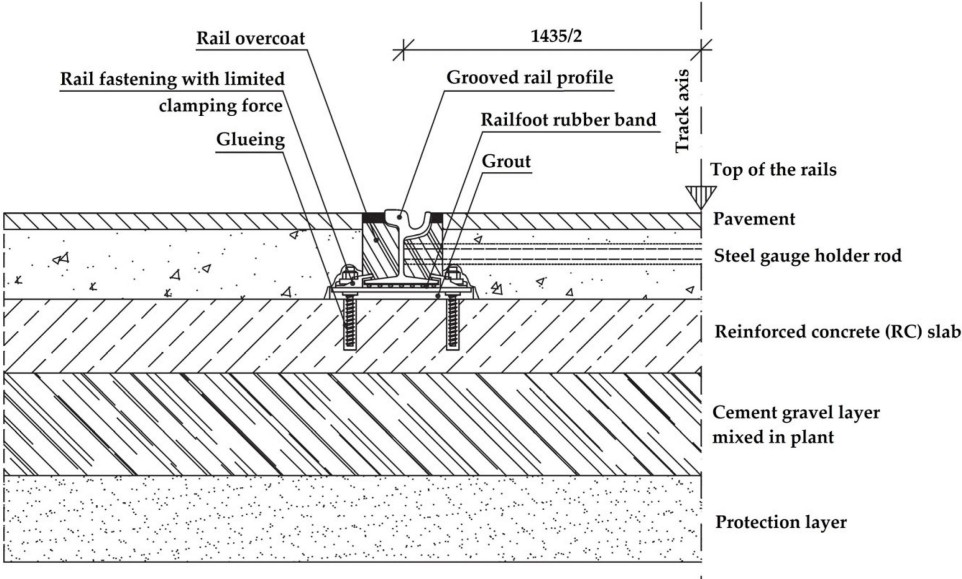

**Figure 1.** Cross-section of the ESCRB I track system (formatted based on [60]).

In the case of the ESCRB II superstructure system (Figure 2), steel fastenings are not installed. Similar to the ESCRB I system, the superstructure is built with rail overcoats and rail foot rubber bands. Any type of rail system can be used, but, in practice, grooved rails are used, which rest within a reinforced concrete slab. In Budapest, basalt concrete is used in most cases, which also provides supplementary bracing for the track.

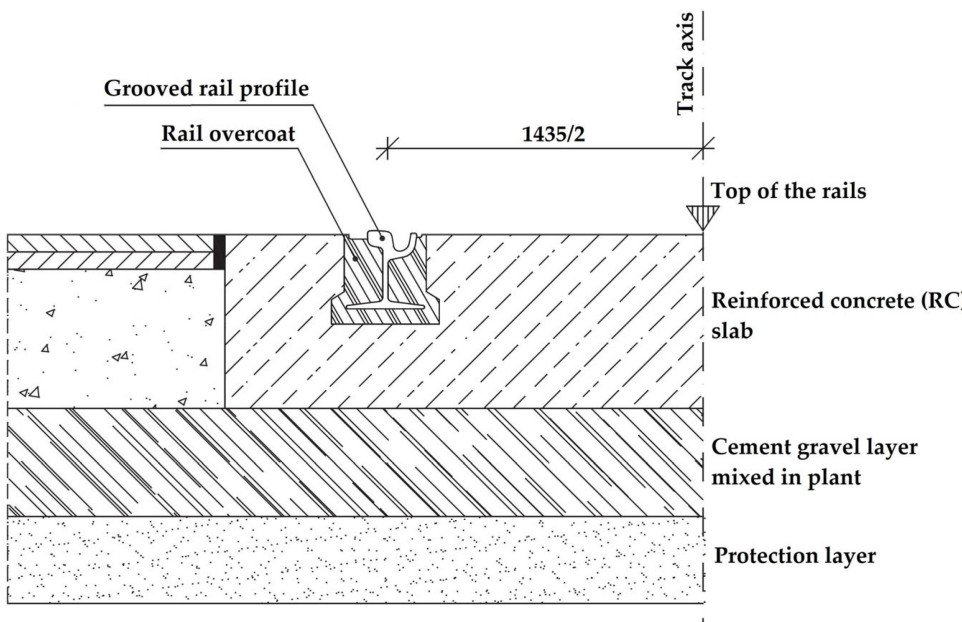

**Figure 2.** Cross-section of the ESCRB II track system (formatted based on [60]).

The third type of ESCRB superstructure system is ESCRB III (Figure 3). The track rests on a reinforced concrete slab or a reinforced concrete beam. Similar to the ESCRB II system, steel fastenings are not installed; instead, there is a homogenous continual elastic support along the entire length of the rail. Nowadays, the use of "filling" technology is very common in Budapest, especially in the case of extremely heavily loaded lines.

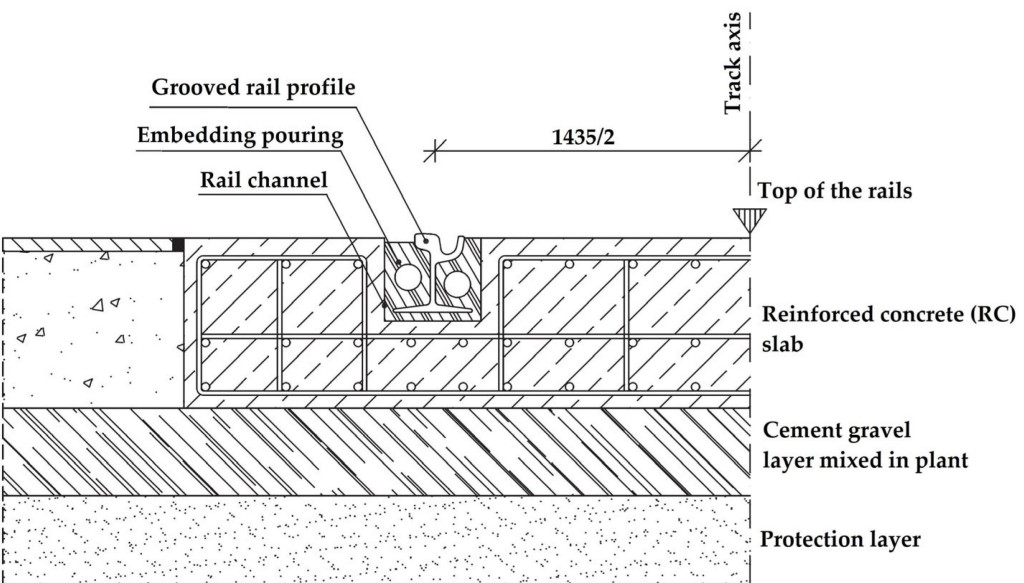

**Figure 3.** Cross-section of ESCRB III track system (formatted based on [60]).

Table 1 shows the general characteristics, while Table 2 represents the geometric characteristics of the reference track sections (i.e., Sections #1 to #11); additionally, Figures 4 and A1–A10 (in Appendix A) show representative photos of all the sections. Figure 5 represents a map of the tramway network of Budapest.

**Table 1.** General characteristics of the reference sections.

| ID. of Reference Sections | Type of Superstructure System | Year of Construction | Details of Superstructure System | Number of Level Crossings |
|---|---|---|---|---|
| Section #1 | ESCRB I | 2008 | 59R2 grooved rails, Gantry rail fastening on a reinforced concrete slab | 6 |
| Section #2 | ESCRB I | 2003 | 59R2 grooved rails, Gantry rail fastening on a reinforced concrete slab | 10 |
| Section #3 | ESCRB I | 2014 | 59R2 grooved rails, Gantry rail fastening on a reinforced concrete slab | 3 |
| Section #4 | ESCRB II | 2011 | 59R2 grooved rails, no steel fastenings, in a reinforced concrete slab | 1 |
| Section #5 | ESCRB II | 2014 | 59R2 grooved rails, no steel fastenings, in a reinforced concrete slab | 3 |
| Section #6 | ESCRB II | 2016 | 59R2 grooved rails, no steel fastenings, in a reinforced concrete slab | 5 |
| Section #7 | ESCRB II | 2014 | 59R2 grooved rails, no steel fastenings, in a reinforced concrete slab | 6 |
| Section #8 | ESCRB III | 2001 | 51R1 grooved rails, embedded with homogenous continual elastic support, in a reinforced concrete overbridge | 0 |
| Section #9 | ESCRB III | 2009 | 53R1 grooved rails, embedded with homogenous continual elastic support, in a reinforced concrete slab | 0 |
| Section #10 | ESCRB III | 2014 | 51R1 grooved rails, embedded with homogenous continual elastic support, in a reinforced concrete overbridge | 0 |
| Section #11 | ESCRB III | 2018 | 59R2 grooved rails, embedded with homogenous continual elastic support, in a reinforced concrete slab | 2 |

**Table 2.** Geometric characteristics of reference sections.

| ID of Reference Sections | Alignment | Most Considerable Curve L (Length) (in m) R (Radius) (in m) | Smallest Curve L (Length) (in m) R (Radius) (in m) |
|---|---|---|---|
| Section #1 | mainly straight, but there are several curves | L = 89 m R = 3000 m | L = 68 m R = 100 m |
| Section #2 | mainly straight | L = 127 m R = 1500 m | L = 50 m R = 400 m |
| Section #3 | mainly straight, but there are several curves | L = 74 m R = 2000 m | L = 61 m R = 450 m |
| Section #4 | straight sections and curves replace each other | L = 40 m R = 1150 m | L = 46 m R = 78 m |
| Section #5 | straight, with three curves | L = 51 m R = 550 m | L = 14 m R = 185 m |
| Section #6 | straight sections and curves replace each other | L = 85 m R = 403 m | L = 40 m R = 103 m |
| Section #7 | straight sections and curves replace each other | L = 26 m R = 1200 m | L = 31 m R = 42 m |
| Section #8 | straight | — | — |
| Section #9 | straight | — | — |
| Section #10 | straight | — | — |
| Section #11 | mainly straight | L = 53 m R = 250 m | — |

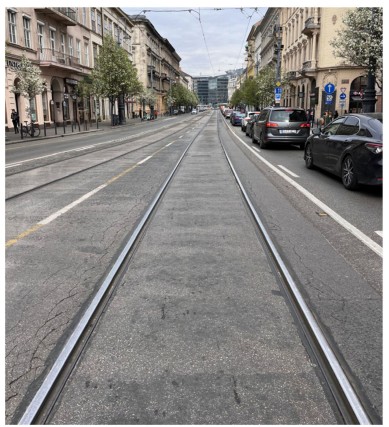

**Figure 4.** Photo of Section #1.

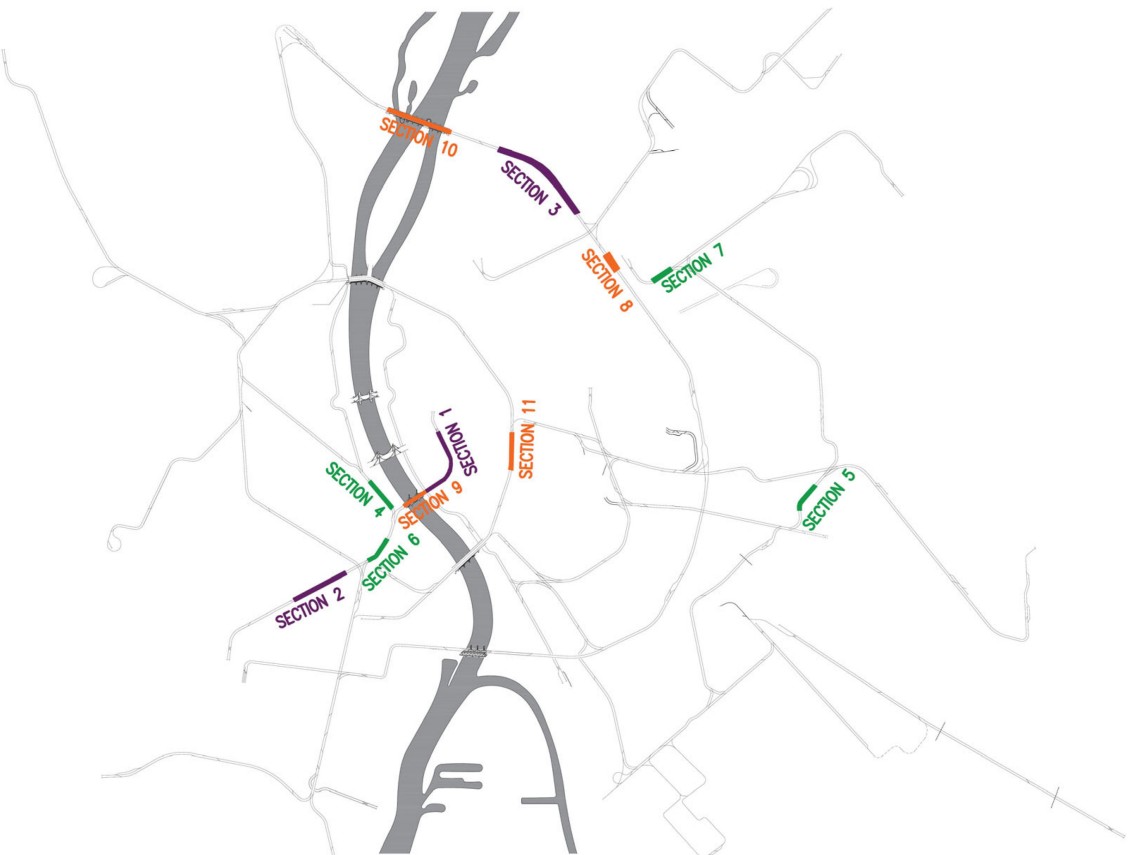

**Figure 5.** Tramway network of Budapest (the different analyzed sections are drawn with own labels, which can be identified by Tables 1 and 2).

### 2.2. *Traffic Load and Age*

In Hungary, the annual through-rolled tonnage values are the most widely used characterization for defining the traffic load of a section. The through-rolled axle tonnage is the mass of all crossing vehicles on a specific line, traveling in one direction in a given year. The value is determined by multiplying the total number of crossing vehicles on the line by the average T0 loading (a vehicle in service, without a crew or passengers) and T3 loading (a serviceable vehicle, with its staff and the maximum passenger capacity) [1,58,60,61]. On the basis of these values, four traffic load classifications can be distinguished (Table 3). This table has also been published in Ref. [1].

**Table 3.** Traffic load classes in the case of Budapest's tramway lines (based on [60].)

| Traffic Load Class | MGT [1]/Year/Direction |
|---|---|
| I./A. extremely heavily loaded line | >7.5 |
| I./B. heavily loaded line | 5.0–7.5 |
| II. medium-loaded line | 2.5–5.0 |
| III. low-loaded line | <2.5 |

[1] MGT indicates million gross tons, i.e., the through-rolled tonnages.

In addition to the traffic load, the reference sections' ages must be considered. Each section's age is precisely known, by year; in this article, the section ages are expressed relative to 2023. The age and average traffic load of the paved reference sections that were investigated between 2017 and 2022 are displayed in Table 4.

**Table 4.** Characteristics of the examined reference sections.

| ID of Reference Sections | Type of Superstructure System | Average Traffic Load (MGT/Year) | Age (Year) |
|---|---|---|---|
| Section #1 | ESCRB I | 4.11 | 15 |
| Section #2 | ESCRB I | 3.80 | 20 |
| Section #3 | ESCRB I | 6.89 | 9 |
| Section #4 | ESCRB II | 4.69 | 12 |
| Section #5 | ESCRB II | 5.77 | 9 |
| Section #6 | ESCRB II | 9.14 | 7 |
| Section #7 | ESCRB II | 4.29 | 9 |
| Section #8 | ESCRB III | 6.81 | 22 |
| Section #9 | ESCRB III | 4.11 | 14 |
| Section #10 | ESCRB III | 6.88 | 9 |
| Section #11 | ESCRB III | 12.78 | 5 |

Table 4 shows that these sections were not built that long ago. The reason for this is that the use of these superstructure systems appeared in our country at the beginning of the 21st century. The most heavily loaded reference track section's superstructure type is the ESCRB III track, which is the most modern technology used for constructing tramway tracks.

### 2.3. Geometrical Measurements and the Examined Parameters

The geometrical measurements were executed, on average, every third month during a night standstill, using a TrackScan 4.01 instrument (Figure 6). Unfortunately, because of the cold weather and the dirty conditions of the tracks, there was no chance to take measurements in winter in the case of paved reference sections. For these reasons, the measurements were made in June 2021, September 2021, April 2022, September 2022, and May 2023. The examination cycle and its evaluation cover nearly two years.

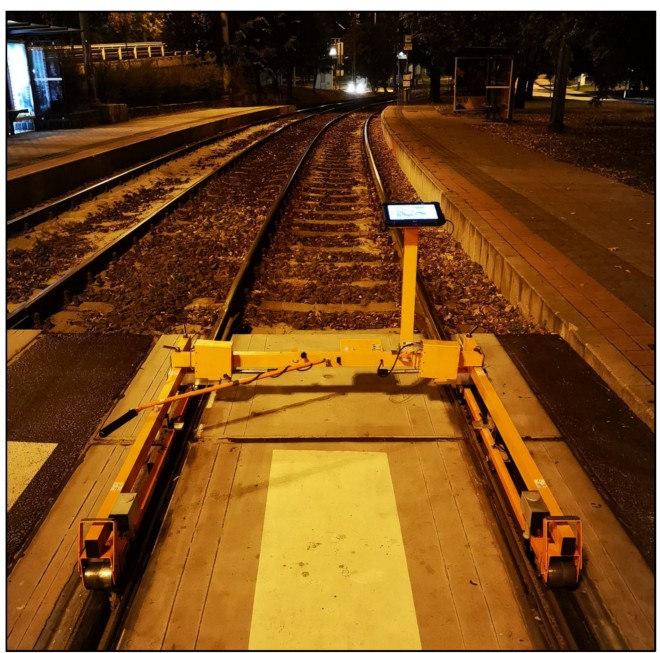

**Figure 6.** TrackScan 4.01 instrument (author's own photo).

The TrackScan 4.01 instrument is a complex track-measuring device; its attributes were described in the authors' earlier article [1]. The parameters measured are the track gauge, alignment, longitudinal level, and twist; however, the twist is not taken into consideration in the current paper. The details of the data processing method are the same as those published in [1].

### 3. Results

First, the results of the geometric analysis are presented separately to analyze and understand the changes in the geometric characteristics of each reference section. Then, in Section 3.12, they will be summed up, considering the traffic load and the sections' ages.

### 3.1. Section #1

Figure 7 shows the changes in the average values of the track gauge parameter. Between the first and the second measurements, the change was almost 0.5 mm, which can be explained by the weather conditions. After a hot summer, this broadening of the track gauge is typical. The same change can be seen between the second and third measurement results; however, in this case, the narrowing of the track gauge appeared. Between the third and fourth and the fourth and fifth measurement results, there is also an increasing change, the value of which is almost as high as the previous value. In the case of changes in the average values of the track gauge parameter of Section #1, the weather conditions significantly affect the results.

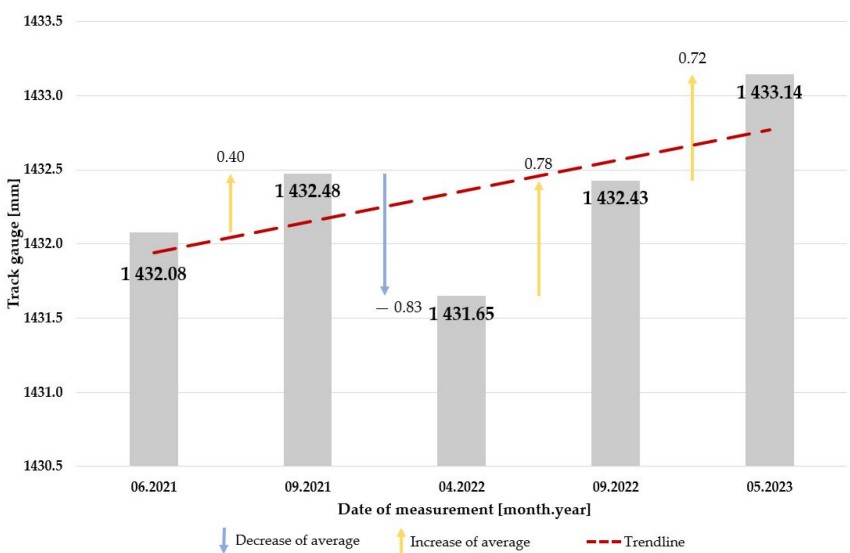

**Figure 7.** The changes in average values of the track gauge in Section #1 (ESCRB I).

Based on the trendline of the measurement data, there is an unequivocal and moderate rise in the Section's track gauge data, which indicates a long-term widening of the track gauge.

Along with the change in track gauge parameters, the changes in alignment and longitudinal level parameters were examined. Figure A11 shows that the trendline of the measurement data of alignment is decreasing evenly. As shown in the diagram, the results of the first, the fourth, and the fifth measurements are similar, but the second and third measurements are outstanding, resulting in a decreasing straight trendline. The trendline of the longitudinal level parameter evidently increases.

The conclusions drawn from the changes in the geometric characteristics in the case of Section #1 are the following:

- The trendline of the track gauge parameter is increasing;
- The trendline of the alignment parameter is decreasing;
- The trendline of the longitudinal level is increasing.

### 3.2. Section #2

Section #1 and Section #2 have had very similar traffic loads through the years and their average value differs by only a few tenths, but Section #2 is five years older than Section #1. Due to the similarity, it was expected that the trend lines would also be similar.

Figure A12 illustrates this similarity; there is also an increasing change, which indicates the broadening of the track gauge. It is steeper than the trendline of Section #1, but the type of geometric change is the same.

Based on the results of the track gauge parameter, the same behavior was expected in the case of the trendlines of alignment and longitudinal level parameters. As Figure A13 shows, similar to Section #1, the trendline of alignment is decreasing. In this case, the results of the first, the fourth, and the fifth measurements are similar, as well as the results of the second and third measurements. The trendline of the longitudinal level parameter increases, just as in the case of Section #1.

The conclusions drawn from the changes in the geometric characteristics in the case of Section #2 are the following:

- The trendline of the track gauge parameter is increasing;
- The trendline of the alignment parameter is decreasing;
- The trendline of the longitudinal level is increasing.

### 3.3. Section #3

Section #3 is the youngest of all the selected ESCRB I reference sections, but it has the most considerable traffic load in a year. Since this section is of the same type as the previous ones, based on the above findings, the widening of the track gauge would be expected. In this instance, the trendline of the track gauge parameter shows a distinct increase, indicating the widening of the track gauge is present (Figure A14). The difference between the fourth and fifth measurement results is so significant that it clearly shows an increase in the average of the track gauge values.

Based on the preceding sections, the same behavior was anticipated for the trendlines of the alignment and longitudinal level parameters. Figure A15 displays the findings that are comparable to Sections #1 and #2: the alignment trendline decreases, and the longitudinal level parameter trendline develops.

The conclusions drawn from the changes in the geometric characteristics in the case of Section #3 are the following:

- The trendline of the track gauge parameter is increasing;
- The trendline of the alignment parameter is decreasing;
- The trendline of the longitudinal level is increasing.

### 3.4. Section #4

Section #4 has an ESCRB II superstructure system and is a medium-loaded line. The trendline of the track gauge parameter exhibits a pronounced and sharp rise, indicating a widening of the track gauge (Figure A16). Similar to Section #1, the findings may be explained by the weather. After summer and a warmer spring, such a rise will occur.

Similar to the previously discussed sections, Figure A17 shows the same longitudinal level and alignment trendline. However, the trendline of alignment is decreasing, and the trendline of the longitudinal level parameter increases steeply.

The conclusions drawn from the changes in the geometric characteristics in the case of Section #4 are the following:

- The trendline of the track gauge parameter is increasing;
- The trendline of the alignment parameter is decreasing;
- The trendline of the longitudinal level is increasing.

### 3.5. Section #5

Section #5 has a higher traffic load in a year than Section #4, but it is younger. Figure A18 shows the trendline of the track gauge parameter, which shows an unequivocal increase, i.e., indicating a widening of the track gauge. This is similar to the previous results for Section #4.

As Figure A19 shows, the trendline of the longitudinal level and alignment are the same as for the previous sections. The trendline of the alignment decreases, the trendline of the longitudinal level parameter increases, and their slopes are also similar.

The conclusions drawn from the changes in the geometric characteristics in the case of Section #5 are the following:

- The trendline of the track gauge parameter is increasing;
- The trendline of the alignment parameter is decreasing;
- The trendline of the longitudinal level is increasing.

### 3.6. Section #6

Section #6 has the highest traffic load in a year of all the selected ESCRB II reference sections. It is an extremely heavily loaded line; furthermore, buses also use the track, which means an even greater load. Despite these facts, contrary to expectations, the change in the average values of the track gauge parameters has been minimal over the years. Figure A20 illustrates this; the effects of the weather conditions are obviously very similar for September 2021 and September 2022. The trendline of the track gauge parameter shows a slight increase. In the time between the fourth and fifth measurements, there was a warmer spring, and several buses began to use the section; therefore, the load they cause and the load from tramway vehicles can explain the increase seen in the average value of the track gauge parameter to this level.

The change in the average values of longitudinal level and alignment does not differ from those of the previous sections (Figure A21). The trendline of alignment decreases. The results of the first, the fourth, and the fifth measurements are similar, as well as the results of the second and third measurements, and so this causes a decrease. The trendline of the longitudinal level parameter increases, just as is the case for all the reference sections.

The conclusions drawn from the changes in the geometric characteristics in the case of Section #6 are the following:

- The trendline of the track gauge parameter is increasing;
- The trendline of the alignment parameter is decreasing;
- The trendline of the longitudinal level is increasing.

### 3.7. Section #7

Section #7 is the same age as Section #5 but has less annual traffic volume. Its traffic volume is comparable to that of Section #4. On this basis, it was anticipated that the findings would be comparable to those of Section #4. This hypothesis was confirmed; the trendline of the track gauge parameter shows a pronounced and sharp rise, indicating a widening of the track gauge (Figure A22). Similar to Section #4, weather circumstances may explain the outcomes. The slope of the trendline is almost identical to that of Section #4's trendlines, in terms of the direction of changes (decrease of increase) between individual measurements.

Figure A23 shows the same trendline as the previous sections, where the trendline of the alignment parameter decreases; on the other hand, the trendline of the longitudinal level parameter increases, just as is the case for all the reference sections.

The conclusions drawn from the changes in the geometric characteristics in the case of Section #7 are the following:

- The trendline of the track gauge parameter is increasing;
- The trendline of the alignment parameter is decreasing;
- The trendline of the longitudinal level is increasing.

### 3.8. Section #8

Section #8 is the oldest of the selected reference sections of the ESCRB III superstructure system; it is also a heavily loaded line. The trendline of the track gauge parameter indicates a clear drop, indicating a decreasing track gauge (Figure A24). In this case, the weather conditions cannot explain the changes since the change here is almost the opposite.

In Figure A25, the results are unchanged. The figure shows the decreasing trendline of the alignment parameter and the increasing trendline of the longitudinal level parameter.

The conclusions drawn from the changes in the geometric characteristics in the case of Section #8 are the following:

- The trendline of the track gauge parameter is decreasing;
- The trendline of the alignment parameter is decreasing;
- The trendline of the longitudinal level is increasing.

### 3.9. Section #9

Section #9 has been located on the Liberty Bridge since 2009; it is a medium-loaded line, and only one type of vehicle can run on here. Figure A26 shows that the trendline of the track gauge parameter increases, which is the exact opposite of the results for Section #8. Again, the broadening of the track gauge is evident.

Figure A27 shows the expected results of the decreasing trendline of the alignment parameter and the increasing trendline of the longitudinal level parameter.

The conclusions drawn from the changes in the geometric characteristics in the case of Section #9 are the following:

- The trendline of the track gauge parameter is increasing;
- The trendline of the alignment parameter is decreasing;
- The trendline of the longitudinal level is increasing.

### 3.10. Section #10

The traffic load of Section #10 is very similar to Section #8, but the section is 13 years younger. Based on the traffic load data, it was expected that there would be similar results to those of Section #8. The changes in measurement data are not identical, but the trendline of the track gauge demonstrates an unambiguous fall, indicating the narrowing of the track gauge (Figure A28).

As Figure A29 shows, the longitudinal level and alignment trendline are the same as the previous ones. The trendline of alignment decreases, and the trendline of the longitudinal level parameter increases; its slope is also similar to the previous ones.

The conclusions drawn from the changes in the geometric characteristics in the case of Section #10 are the following:

- The trendline of the track gauge parameter is decreasing;
- The trendline of the alignment parameter is decreasing;
- The trendline of the longitudinal level is increasing.

### 3.11. Section #11

The traffic load of Section #11 is the highest of all the selected reference sections; it is an extremely heavily loaded line in Budapest. Siemens Combino NF12B vehicles run on this line every five minutes, even at night, with an exceptional axis load. Section #11 is from Budapest's busiest and most frequently used line; this reference section was built in 2018.

Despite this considerable traffic load, when considering the last two years, the change in the average of the track gauge parameter is very minimal overall and does not follow the pattern of change due to weather conditions. Figure A30 shows an increasing change, which indicates the broadening of the track gauge.

Figure A31 shows the same pattern: the trendline of the measurement data of alignment decreases. However, the trendline of the longitudinal level parameter evidently increases.

The conclusions drawn from the changes in the geometric characteristics in the case of Section #11 are the following:

- The trendline of the track gauge parameter is increasing;
- The trendline of the alignment parameter is decreasing;
- The trendline of the longitudinal level is decreasing.

### 3.12. Results for All Superstructure Systems

Table 5 summarizes the changes in the trendlines of the average values of the geometric parameters of the paved superstructure systems.

**Table 5.** Summary of changes in the trendlines of the average values of the geometric parameters of the paved superstructure systems.

| Type of Superstructure System | Traffic Load Class | Change in Track Gauge Parameter | Change in Alignment Parameter | Change in Longitudinal Parameter |
|---|---|---|---|---|
| ESCRB I | medium-loaded line | increasing | decreasing | increasing |
| ESCRB I | heavily loaded line | increasing | decreasing | increasing |
| ESCRB I | extremely heavily loaded line | N/A | N/A | N/A |
| ESCRB II | medium-loaded line | increasing | decreasing | increasing |
| ESCRB II | heavily loaded line | increasing | decreasing | increasing |
| ESCRB II | extremely heavily loaded line | increasing | decreasing | increasing |
| ESCRB III | medium-loaded line | increasing | decreasing | increasing |
| ESCRB III | heavily loaded line | decreasing | decreasing | increasing |
| ESCRB III | extremely heavily loaded line | increasing | decreasing | increasing |

## 4. Discussion

In this section, the findings from the geometric analysis of each reference section are reviewed, taking into account the traffic load and the ages of the sections, based on each superstructure system.

### 4.1. Results for the ESCRB I Track System

Figure 8 depicts the link between the trendlines of track gauge modifications and the average traffic load values for the analyzed ESCRB I track system.

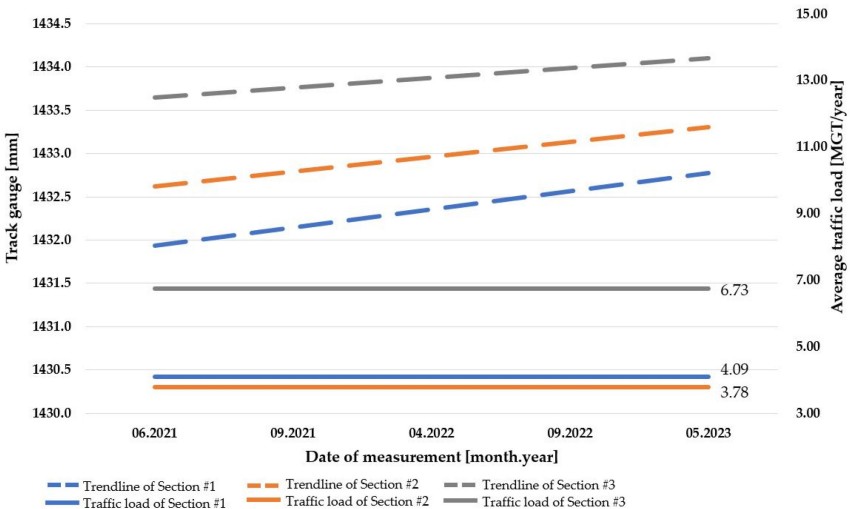

**Figure 8.** The relationship between track gauge and traffic load in the ESCRB I track system.

Section #1 and Section #2 have almost the same average traffic load in a year, but Section #2 is five years older than Section #1. Both have a similar trendline with almost the same steepness.

Section #3 is the youngest ESCRB I track and has more than one and a half times the traffic load of Section #1 and Section #2. Therefore, it is to be expected that in this case as well, the trendline increases more steeply: an evaluation of the results supports this suggestion.

Figure 9 depicts the link between the trendlines of the alignment adjustments for the ESCRB I track system and the average traffic load values.

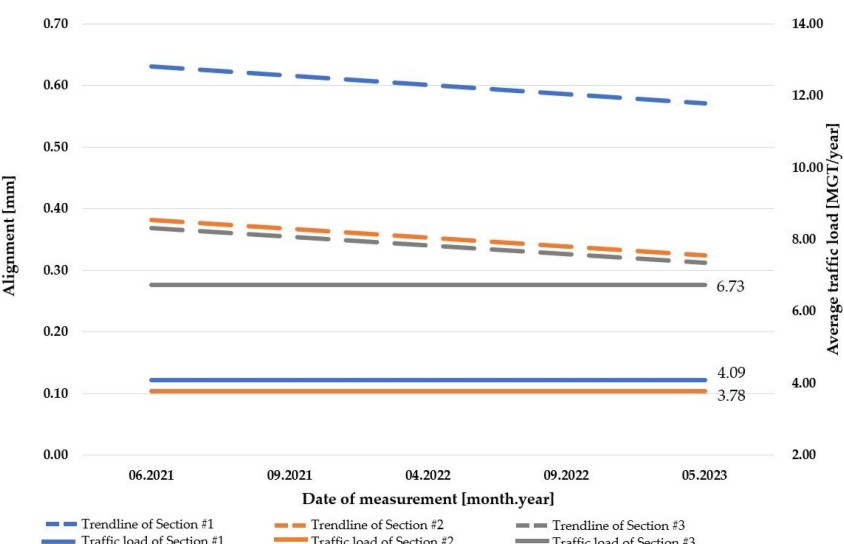

**Figure 9.** The relationship between alignment and traffic load in the ESCRB I track system.

Regardless of the traffic load level, each reference section's trendlines decrease. It is interesting that regardless of the age or the traffic load of the selected reference section, the results of the first, the fourth, and the fifth measurements are similar, as well as the results of the second and the third measurements, resulting in a decreasing trendline.

Figure 10 demonstrates the correlation between the trendlines of longitudinal level changes for the evaluated ESCRB I track system and average traffic load levels.

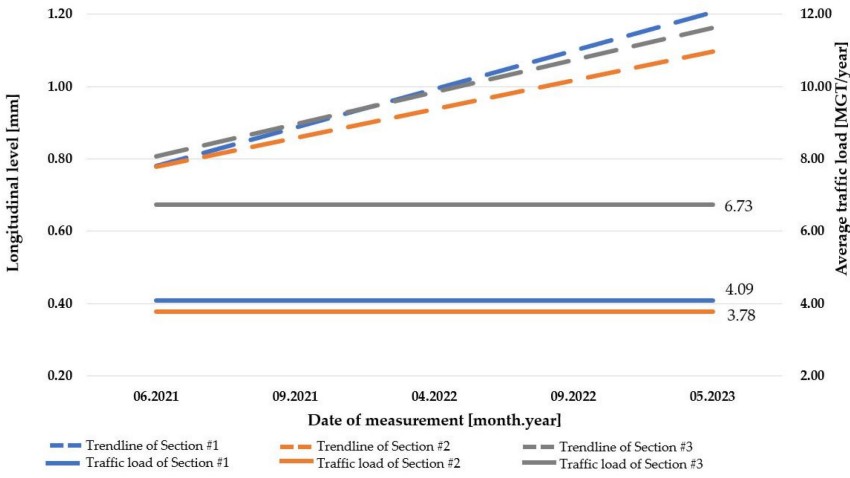

**Figure 10.** The relationship between longitudinal level and traffic load in the ESCRB I track system.

Regardless of the traffic volume or age of the chosen reference segment, the trendlines rise sharply. Section #1, which is middle-aged and has a medium traffic load value, might show a steeper trendline than the others. In general, these values are constantly increasing from year to year.

Based on the results, the following statements can be made about the ESCRB I tracks:

- Regardless of traffic load or age, the track gauge trendline is growing, indicating a widening track gauge, and similar degradation might be seen;
- In the case of older reference sections, weather conditions have an evident impact on the findings;
- The average value of alignment is decreasing, regardless of traffic load or age, and the degradation is comparable;
- Despite traffic load or age, the average value of the longitudinal level increases annually, and the degradation is comparable.

Tables 6–8 show a summary of the findings from the geometric analysis of the reference sections from the ESCRB I superstructure system.

**Table 6.** Summary of the geometric analysis of the track gauge parameter in the case of the ESCRB I superstructure system.

| ID of Reference Sections | Average Traffic Load (MGT/Year) | Age (Year) | Change in Track Gauge Parameter | Summary |
|---|---|---|---|---|
| Section #1 | 4.11 | 15 | increasing | similar degradation, regardless |
| Section #2 | 3.80 | 20 | increasing | of traffic load or age (the track |
| Section #3 | 6.89 | 9 | increasing | gauge is widening) |

**Table 7.** Summary of the geometric analysis of the alignment parameter in the case of the ESCRB I superstructure system.

| ID of Reference Sections | Average Traffic Load (MGT/Year) | Age (Year) | Change in Alignment Parameter | Summary |
|---|---|---|---|---|
| Section #1 | 4.11 | 15 | decreasing | the trendline is consistently |
| Section #2 | 3.80 | 20 | decreasing | decreasing, regardless of traffic |
| Section #3 | 6.89 | 9 | decreasing | load or age |

**Table 8.** Summary of the geometric analysis of the longitudinal level parameter in the case of the ESCRB I superstructure system.

| ID of Reference Sections | Average Traffic Load (MGT/Year) | Age (Year) | Change in Longitudinal Level Parameter | Summary |
|---|---|---|---|---|
| Section #1 | 4.11 | 15 | increasing | the trendline is consistently |
| Section #2 | 3.80 | 20 | increasing | increasing, regardless of traffic |
| Section #3 | 6.89 | 9 | increasing | load or age |

*4.2. Results for the ESCRB II Track System*

Figure 11 demonstrates the connection between the trendlines of track gauge modifications and the average traffic load values for the ESCRB II track system being evaluated.

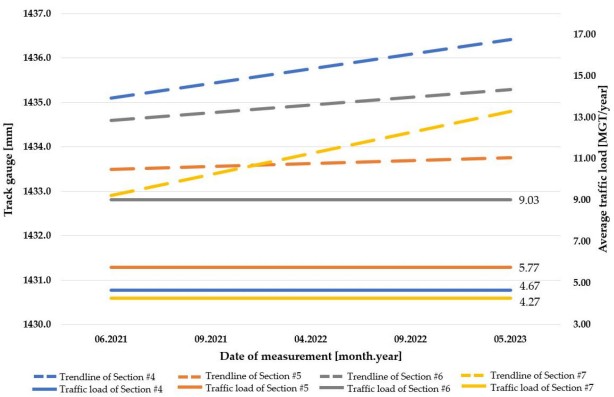

**Figure 11.** The relationship between track gauge and traffic load in the ESCRB II track system.

Section #4 and Section #7 have almost the same average traffic load in a year, and Section #7 is only three years older than Section #4. As a result, both have similarly steep trendlines, which indicate the broadening of the track gauge through the years.

Section #5 and Section #6 have a slightly higher average traffic load; the change in these cases is also an increase, which also indicates the broadening of the track gauge through the years.

Figure 12 displays the relation between the trendlines of the alignment variations and the average traffic load values for the ESCRB II track system.

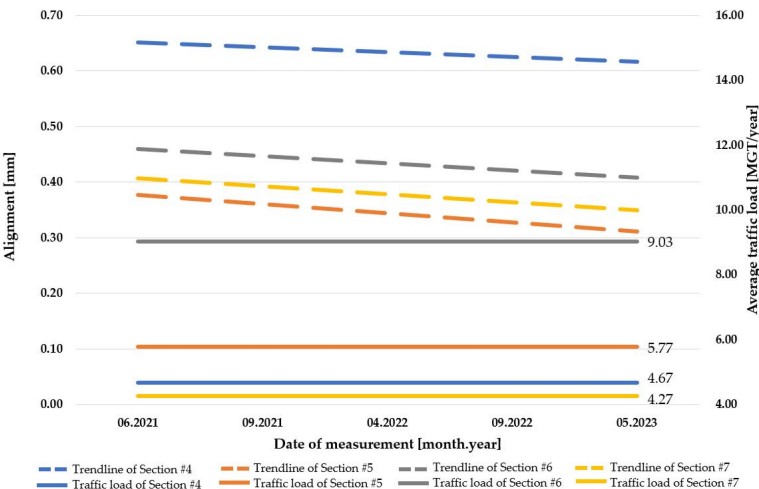

**Figure 12.** The relationship between alignment and traffic load in the ESCRB II track system.

For each reference section in Figure 12, the trendlines decrease. It is the same scenario as for the ESCRB I tracks; regardless of the age or traffic load of the selected reference section, the results of the first, the fourth, and the fifth measurements are similar, as well as the results of the second and third measurements.

Figure 13 shows the connection between the longitudinal level change trendlines and average traffic load values for the evaluated ESCRB II track system. In every case, the trendlines are steeply increasing.

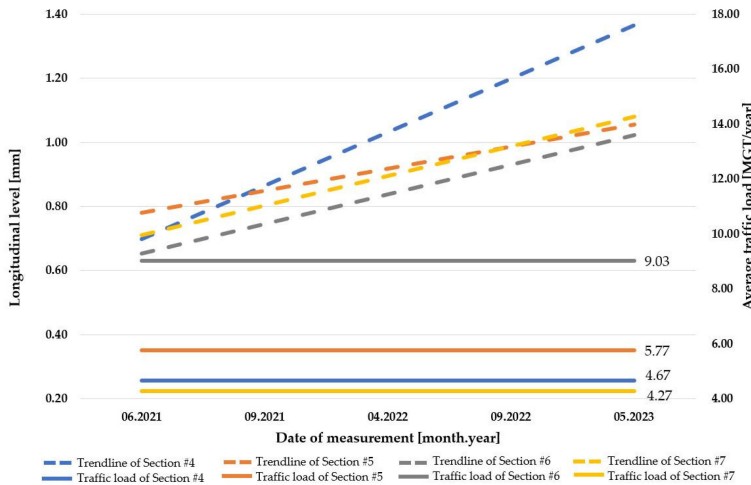

**Figure 13.** The relationship between longitudinal level and traffic load in the ESCRB II track system.

According to the findings, the following can be stated regarding the ESCRB II tracks:

- Regardless of traffic load or age, the track gauge trendline is growing, indicating a widening track gauge, where similar degradation might be seen;
- When the ESCRB II track carries less traffic load than another ESCRB II track, the trendline of the track gauge parameter is steeper, so that the widening of the track gauge is faster;
- Notwithstanding the greater age of the section, the weather conditions significantly influence the findings;
- The average value of alignment is decreasing, regardless of traffic load or age, and the degradation is comparable;

- The longitudinal level's average value is consistently growing from year to year, regardless of traffic volume or age, and the degradation is comparable.

Tables 9–11 show a summary of the findings from the geometric analysis of the reference sections from the ESCRB II superstructure system.

**Table 9.** Summary of the geometric analysis of the track gauge parameter in the case of the ESCRB II superstructure system.

| ID of Reference Sections | Average Traffic Load (MGT/Year) | Age (Year) | Change in Track Gauge Parameter | Summary |
|---|---|---|---|---|
| Section #4 | 4.69 | 12 | increasing | similar degradation, regardless of traffic load or age (track gauge is widening) |
| Section #5 | 5.77 | 9 | increasing | |
| Section #6 | 9.14 | 7 | increasing | |
| Section #7 | 4.29 | 9 | increasing | |

**Table 10.** Summary of the geometric analysis of the alignment parameter in the case of the ESCRB II superstructure system.

| ID of Reference Sections | Average Traffic Load (MGT/Year) | Age (Year) | Change in Alignment Parameter | Summary |
|---|---|---|---|---|
| Section #4 | 4.69 | 12 | decreasing | the trendline is consistently decreasing, regardless of traffic load or age |
| Section #5 | 5.77 | 9 | decreasing | |
| Section #6 | 9.14 | 7 | decreasing | |
| Section #7 | 4.29 | 9 | decreasing | |

**Table 11.** Summary of the geometric analysis of the longitudinal level parameter in the case of the ESCRB II superstructure system.

| ID of Reference Sections | Average Traffic Load (MGT/Year) | Age (Year) | Change in Longitudinal Level Parameter | Summary |
|---|---|---|---|---|
| Section #4 | 4.69 | 12 | increasing | the trendline is consistently increasing, regardless of traffic load or age |
| Section #5 | 5.77 | 9 | increasing | |
| Section #6 | 9.14 | 7 | increasing | |
| Section #7 | 4.29 | 9 | increasing | |

*4.3. Results for the ESCRB III Track System*

Figure 14 shows the connection between the trendlines of track gauge variations in the examined ESCRB III track system and the average traffic load values.

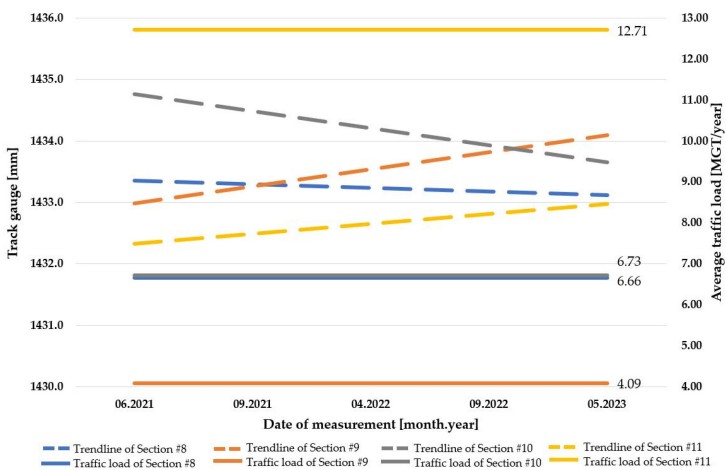

**Figure 14.** The relationship between track gauge and traffic load in the ESCRB III track system.

Section #8 and Section #10 have almost the same average traffic load in a year, but Section #8 is thirteen years older than Section #10. Both have similarly steep trendlines, which indicates the narrowing of the track gauge through the years.

Section #9 has the lowest average traffic load in a year, while Section #11 has the highest. Section #11 was built only five years ago, while Section #9 was built 14 years ago. The steepness of the trendlines of Section #9 and Section #11 are similar.

Unfortunately, based on these findings, it is not clear what factors influence the geometric changes in the track gauge parameter; therefore, it is necessary to investigate these factors further. However, what is clear is that Section #11 is the most frequently used line in Budapest, so its deterioration can be expected sooner.

Figure 15 depicts the correlation between the trendlines of the alignment variations and the average traffic load values for the ESCRB III track system.

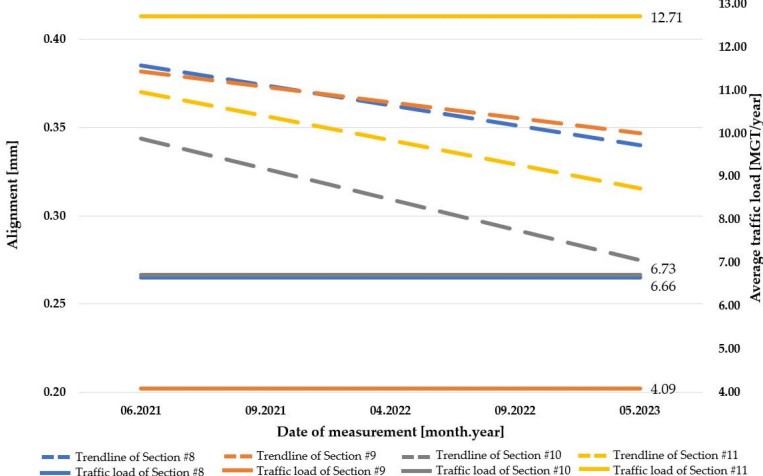

**Figure 15.** The relationship between alignment and traffic load in the ESCRB III track system.

As is the case with the ESCRB II tracks, all the trendlines decrease. It is the same scenario as for the previously discussed paved tracks: regardless of the age or traffic load of the selected reference section, the results of the first, the fourth, and the fifth measurements are similar, as well as the results of the second and third measurements.

Figure 16 displays the connection between the trendlines of the longitudinal level changes for the tested ESCRB III track system and the average traffic load values.

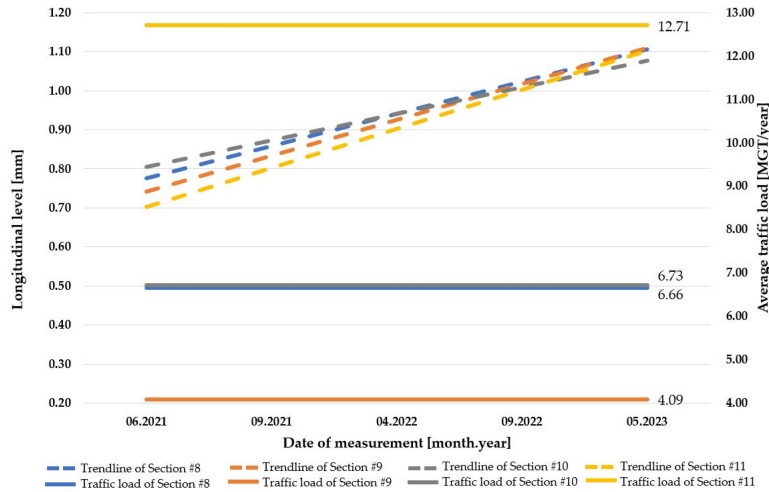

**Figure 16.** The relationship between longitudinal level and traffic load in the ESCRB III track system.

The trendlines are steeply increasing, as is the case with the ESCRB I and ESCRB II tracks. The following can be deduced about the ESCRB III tracks, based on the results:

- When the examined ESCRB III reference section is a medium loaded line, regardless of the age, the trendline of the track gauge is increasing, which means the broadening of the track gauge, and the deterioration is comparable;
- The average value of alignment is decreasing, regardless of traffic load or age, and the degradation is comparable;
- Regardless of traffic volume or age, the annual average value of the longitudinal level is consistently increasing, and the degradation is comparable.

Tables 12–14 show a summary of the findings of the geometric analysis of the reference sections from the ESCRB III superstructure system.

**Table 12.** Summary of the geometric analysis of the track gauge parameter in the case of the ESCRB III superstructure system.

| ID of Reference Sections | Average Traffic Load (MGT/Year) | Age (Year) | Change in Track Gauge Parameter | Summary |
|---|---|---|---|---|
| Section #8 | 6.81 | 22 | decreasing | similar degradation in the case of medium-loaded lines (the track gauge is broadening) |
| Section #9 | 4.11 | 14 | increasing | |
| Section #10 | 6.88 | 9 | decreasing | |
| Section #11 | 12.78 | 5 | increasing | |

**Table 13.** Summary of the geometric analysis of the alignment parameter in the case of the ESCRB III superstructure system.

| ID of Reference Sections | Average Traffic Load (MGT/Year) | Age (Year) | Change in Alignment Parameter | Summary |
|---|---|---|---|---|
| Section #8 | 6.81 | 22 | decreasing | the trendline is consistently decreasing, regardless of traffic load or age |
| Section #9 | 4.11 | 14 | decreasing | |
| Section #10 | 6.88 | 9 | decreasing | |
| Section #11 | 12.78 | 5 | decreasing | |

**Table 14.** Summary of the geometric analysis of the longitudinal level parameter in the case of the ESCRB III superstructure system.

| ID of Reference Sections | Average Traffic Load (MGT/Year) | Age (Year) | Change in Longitudinal Parameter | Summary |
|---|---|---|---|---|
| Section #8 | 6.81 | 22 | increasing | the trendline is consistently increasing, regardless of traffic load or age |
| Section #9 | 4.11 | 14 | increasing | |
| Section #10 | 6.88 | 9 | increasing | |
| Section #11 | 12.78 | 5 | increasing | |

*4.4. Comparison of the Results for the Different Track Systems*

In this subsection, even though further measurements are necessary, the results of the paved superstructure systems are compared.

The following may be stated regarding the examined paved tramway tracks in Budapest, based on the authors' results.

- Regardless of the ESCRB superstructure system or age, the track gauge trendline of a medium-loaded line is growing, indicating a (slight) widening of the track gauge:

  i.   In the case of ESCRB I, the average increase of the trendline is 0.070% in the examined period;

  ii.  In the case of ESCRB II, the average increase of the trendline is 0.144% in the examined period;

  iii. In the case of ESCRB III, the average increase of the trendline is 0.093% in the examined period.

- The annual average value of the longitudinal level trendline continues to rise, independent of traffic volume or age:

  i.   In the case of ESCRB I, the average increase of the trendline is 72.4% in the examined period;

  ii.  In the case of ESCRB II, the average increase of the trendline is 92.9% in the examined period;

  iii. In the case of ESCRB III, the average increase of the trendline is 72.6% in the examined period.

- The annual average value of the alignment trendline continues to descend, independent of traffic volume or age:

  i.   In the case of ESCRB I, the average decrease of the trendline is 4.85% in the examined period;

  ii.  In the case of ESCRB II, the average decrease of the trendline is 1.46% in the examined period;

  iii. In the case of ESCRB III, the average decrease of the trendline is 6.66% in the examined period.

The measures available to counter the deterioration depend on the superstructure system; however, in general, the following maintenance activities can be formulated to slow down the deterioration process.

- Defect in the rail: attachment weld and the grinding or replacement of the rail;
- Defect in the cover: repair or replacement;
- Defect in the rail overcoat: supplement or replacement;
- Defect in the gauge holder rod: replacement;
- Defect in the rail fastening: replacement.

Unfortunately, in Hungary, the most common maintenance work performed is the repair of rail and pavement defects, and these repairs depend on the maintenance department's budget and resources, not necessity.

## 5. Conclusions

Over the years, parallel to the development and transformation of the capital city of Hungary, Budapest, it has become necessary to use new types of superstructure systems in addition to or instead of the ballasted track. In response to these needs, the Budapest Transport Privately Held Corporation, the company responsible for operating and maintaining the railway tracks, began using the paved superstructure systems at the beginning of the 2000s.

Paved tracks have many advantages, but the most important benefits are that maintenance is more manageable and the surface can also be used by road traffic. There are two types of paved superstructure systems: the elastically supported continuous rail bedding system (ESCRB) and the "large" slab superstructure system. The latter is used in road crossings; several ESCRB systems are used in the most heavily loaded lines.

In this study, the authors examine the geometric evolution of many ESCRB superstructure systems. The TrackScan 4.01 device was used to conduct measurements in June 2021, September 2021, April 2022, September 2022, and May 2023. The examined track parameters consisted of track gauge, alignment, and longitudinal level, which were measured and recorded.

The following may be stated regarding the trendlines for the examined paved tramway tracks in Budapest, based on the authors' results:

- Regardless of the ESCRB superstructure system or its age, the track gauge trendline of a medium-loaded line is growing, indicating a (slight) widening of the track gauge;
- The annual average value of the longitudinal level continues to rise, independent of traffic volume or age;
- The annual average value of the alignment continues to descend, independent of traffic volume or age.

Hungary's most common maintenance work depends on budgets and resources, not necessity, even though the deterioration of the tracks could be slowed down with a little regular maintenance. These interventions could be made as follows:

- When the track gauge apparently increases or decreases, it is necessary to dismantle the cover and replacement the gauge holder rod;
- When the rail overcoat deforms due to temperature changes, it is necessary to replace it immediately;
- When the rail is damaged, it is necessary to discover the cause of the fault and remedy it immediately;
- When the cover deforms due to load, it is necessary to fix it immediately.

It should be noted that the diagnostic observation period considered in this paper for the tramway tracks that were analyzed was about 2 years; this will remain so for the time being. The following measurements are planned for September 2023. As mentioned, the present article is a continuation of our previous paper [1] investigating the deterioration of open tramway tracks. Of course, the authors are fully aware that it is impossible to draw general conclusions when considering such a long and extensive deterioration period. On the other hand, it should be noted that no research group in Hungary has been involved in work in this field until now. At the international level, too, the number of scientific publications dealing with this issue is nominal. Therefore, publishing the results in the initial phase of the current research is vital.

The authors want to evaluate the reference sections and assess the necessary modifications in the future. Based on the outcomes, they also plan to investigate how the subsequent measurement results impact the present trend lines. It is also vital to compare the open tramway track to the paved tramway track since the assessments of the data already provide comparable visual results.

**Author Contributions:** Conceptualization, V.J. and S.F.; methodology, V.J. and S.F.; software, V.J. and S.F.; validation, V.J. and S.F.; formal analysis, V.J. and S.F.; investigation, V.J., Z.M., A.N., D.K., M.S. and S.F.; resources, V.J., Z.M., A.N. and S.F.; data curation, V.J. and S.F.; writing—original draft preparation, V.J., Z.M., A.N., D.K., M.S. and S.F.; writing—review and editing V.J., Z.M., A.N., D.K., M.S. and S.F.; visualization, V.J. and S.F.; supervision, Z.M., A.N., D.K., M.S. and S.F.; project administration, V.J. and S.F.; funding acquisition, S.F. All authors have read and agreed to the published version of the manuscript.

**Funding:** This research received no external funding.

**Data Availability Statement:** Not applicable.

**Acknowledgments:** This work was technically supported by BKV PLC. This paper was prepared by the research team "SZE-RAIL".

**Conflicts of Interest:** The authors declare no conflict of interest.

## Appendix A

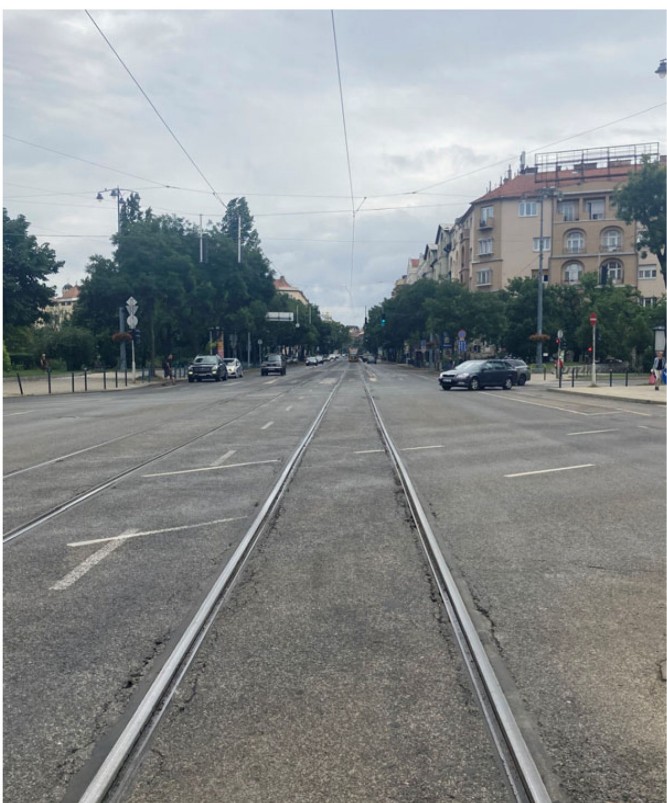

**Figure A1.** Photo of Section #2.

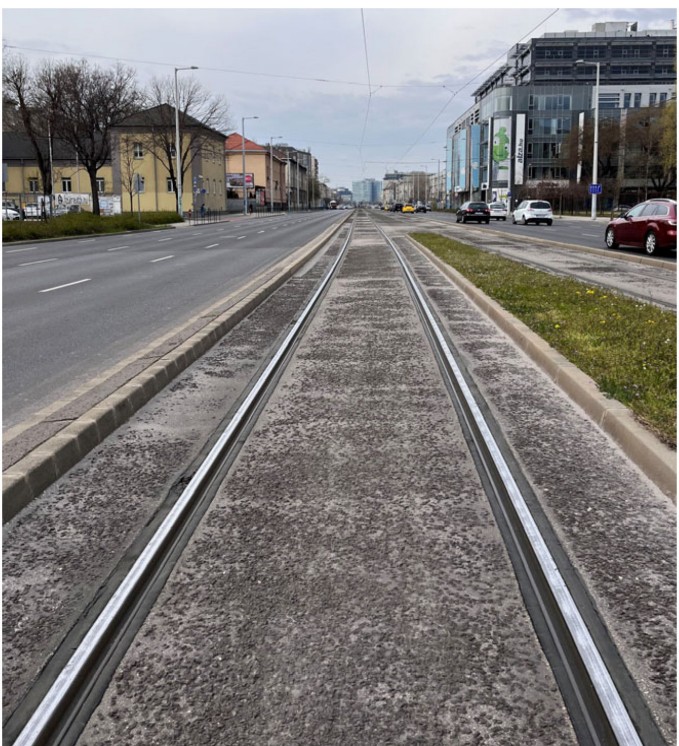

**Figure A2.** Photo of Section #3.

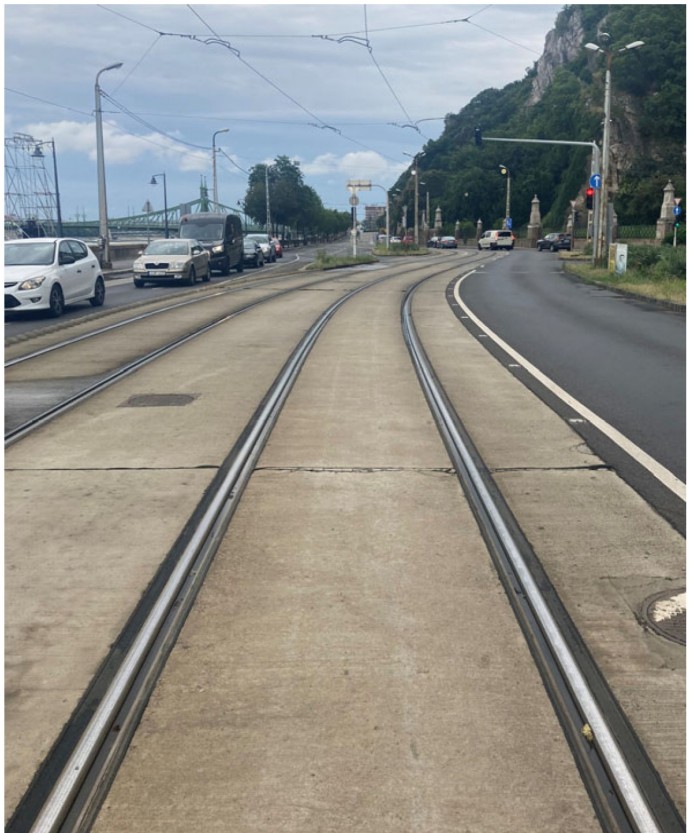

**Figure A3.** Photo of Section #4.

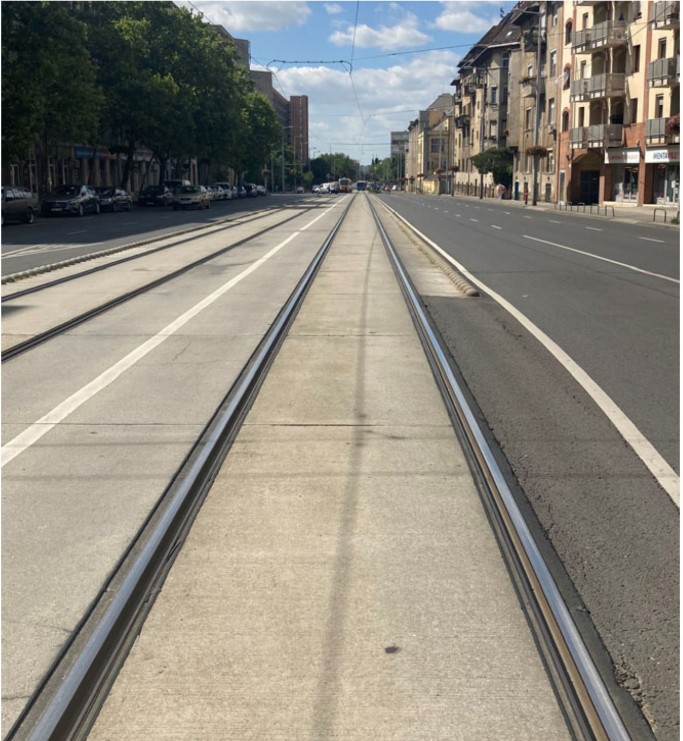

**Figure A4.** Photo of Section #5.

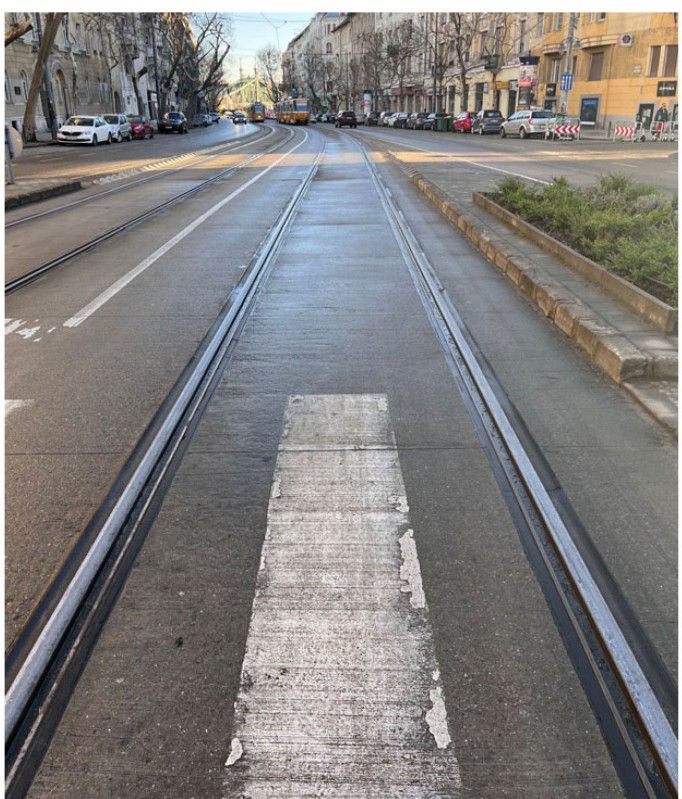

**Figure A5.** Photo of Section #6.

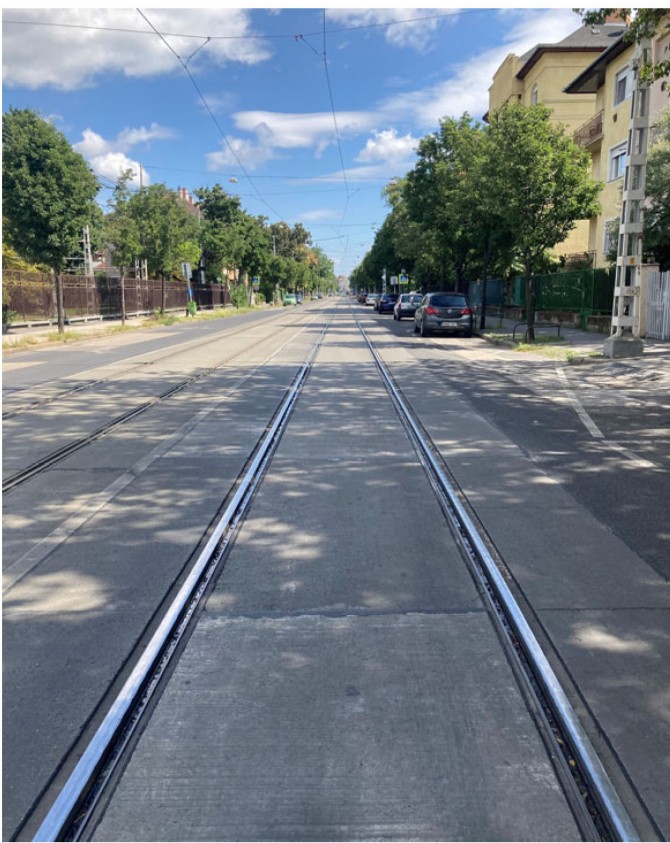

**Figure A6.** Photo of Section #7.

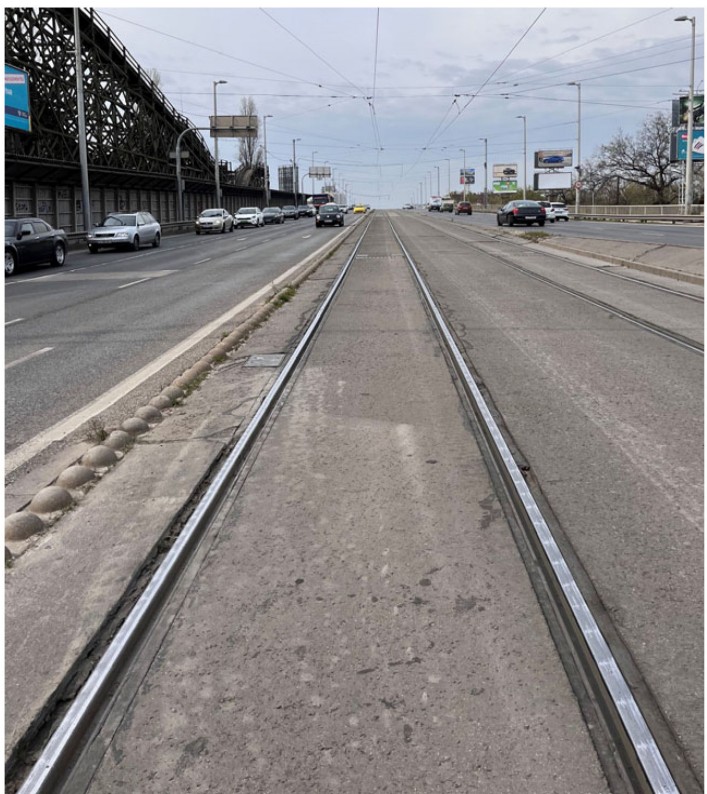

**Figure A7.** Photo of Section #8.

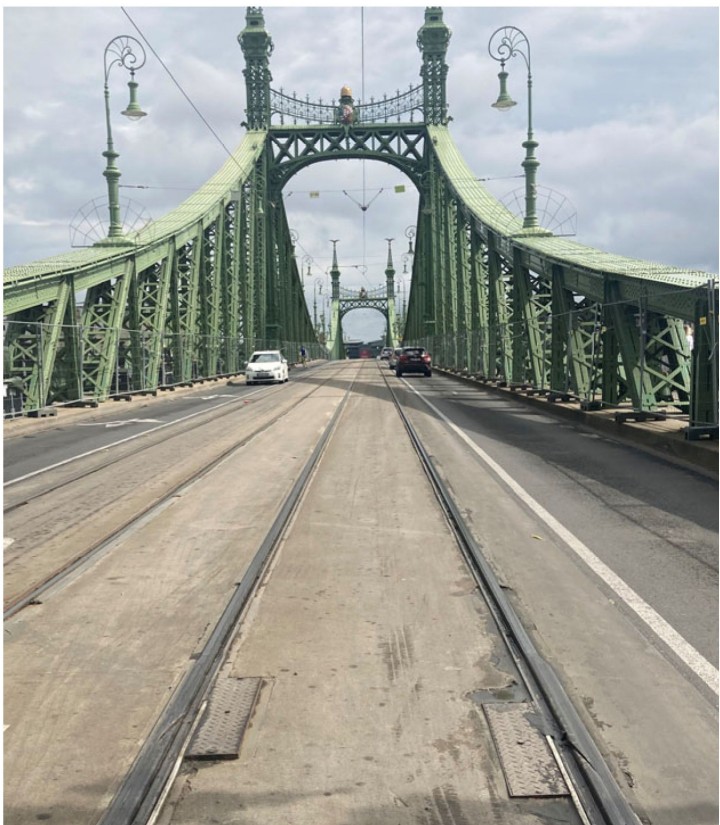

**Figure A8.** Photo of Section #9.

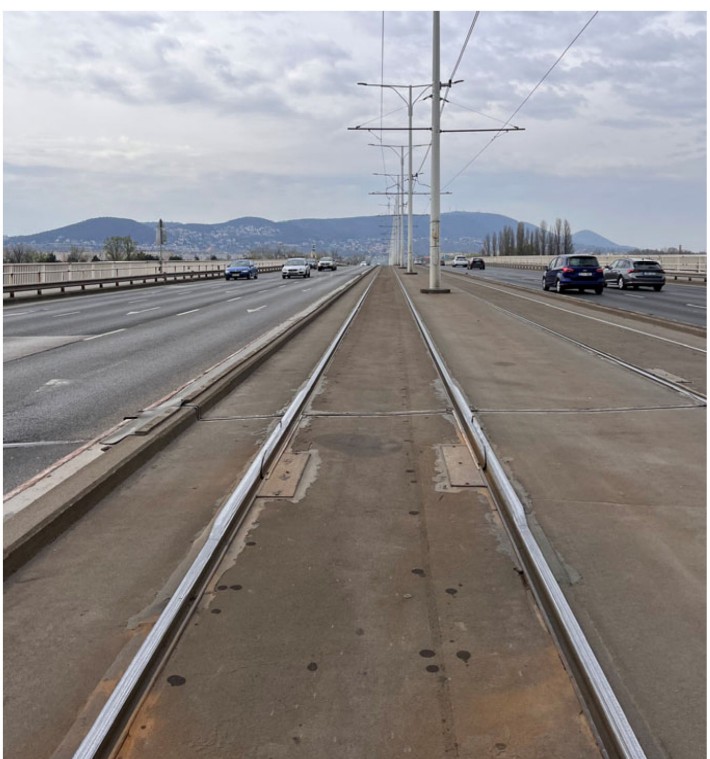

**Figure A9.** Photo of Section #10.

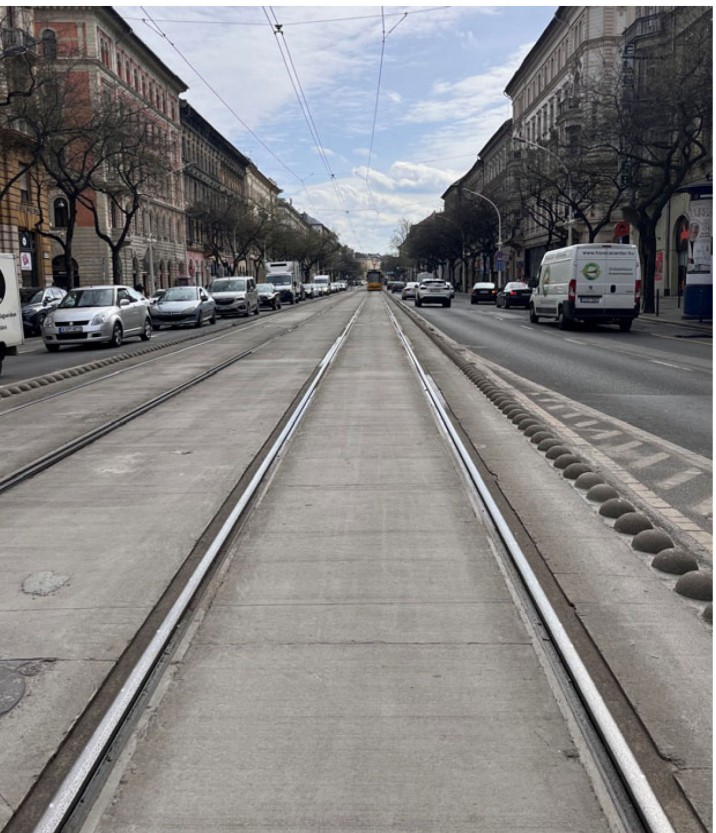

**Figure A10.** Photo of Section #11.

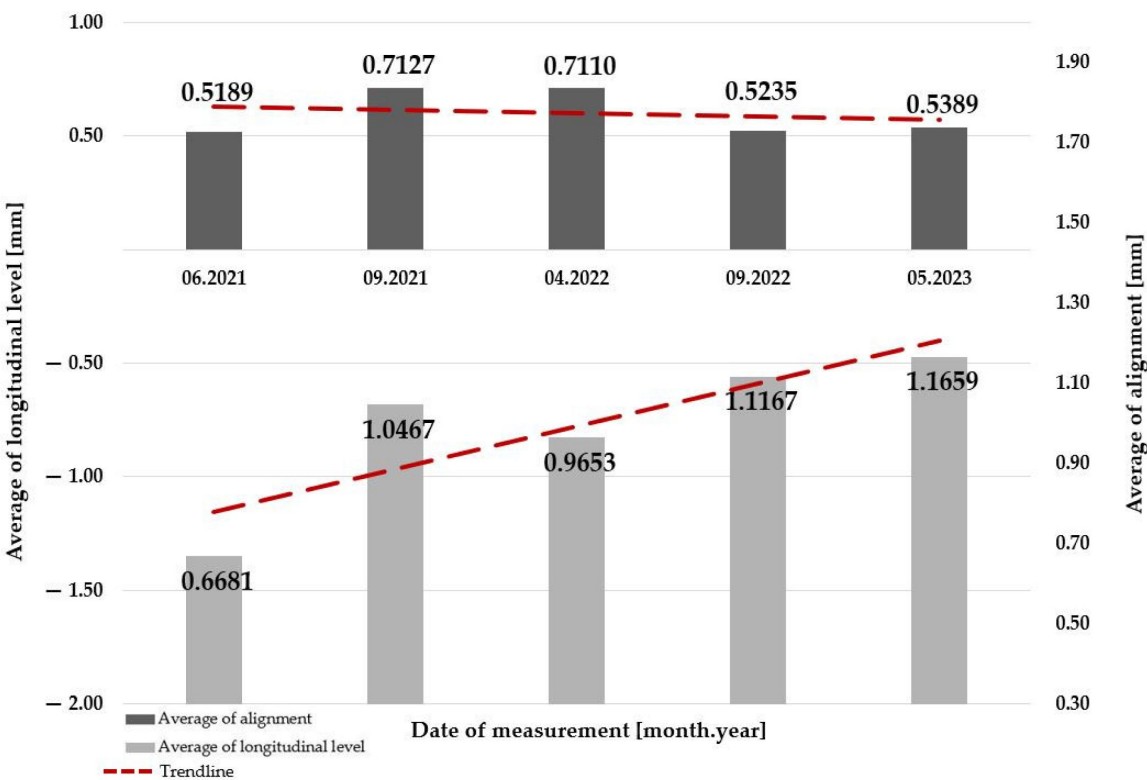

**Figure A11.** The change in the average values for longitudinal level and alignment in Section #1 (ESCRB I).

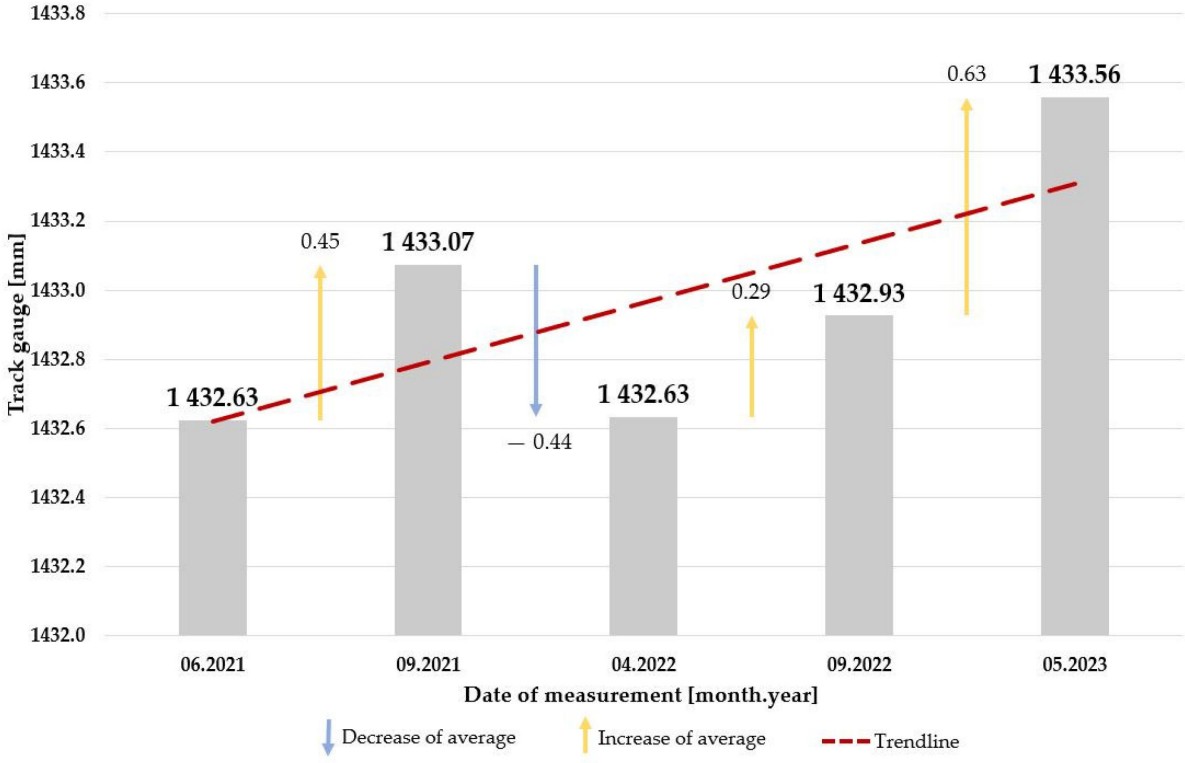

**Figure A12.** The change in the average values for track gauge in Section #2 (ESCRB I).

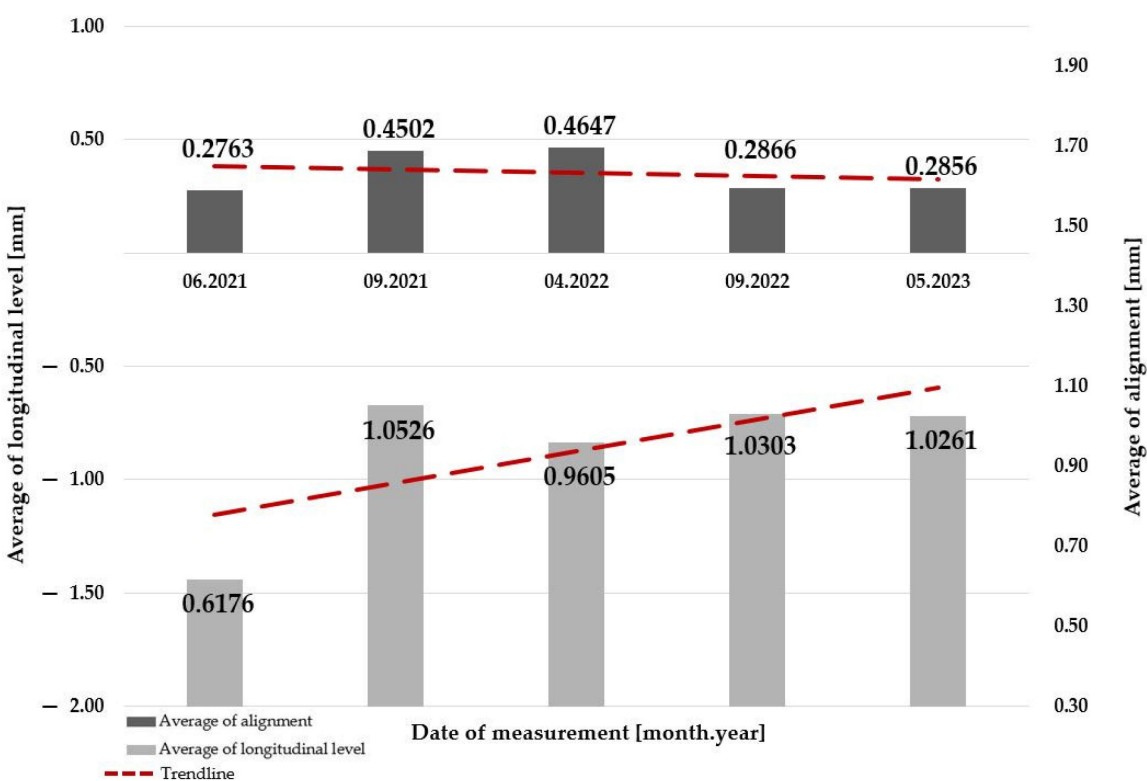

**Figure A13.** The change in the average values for longitudinal level and alignment in Section #2 (ESCRB I).

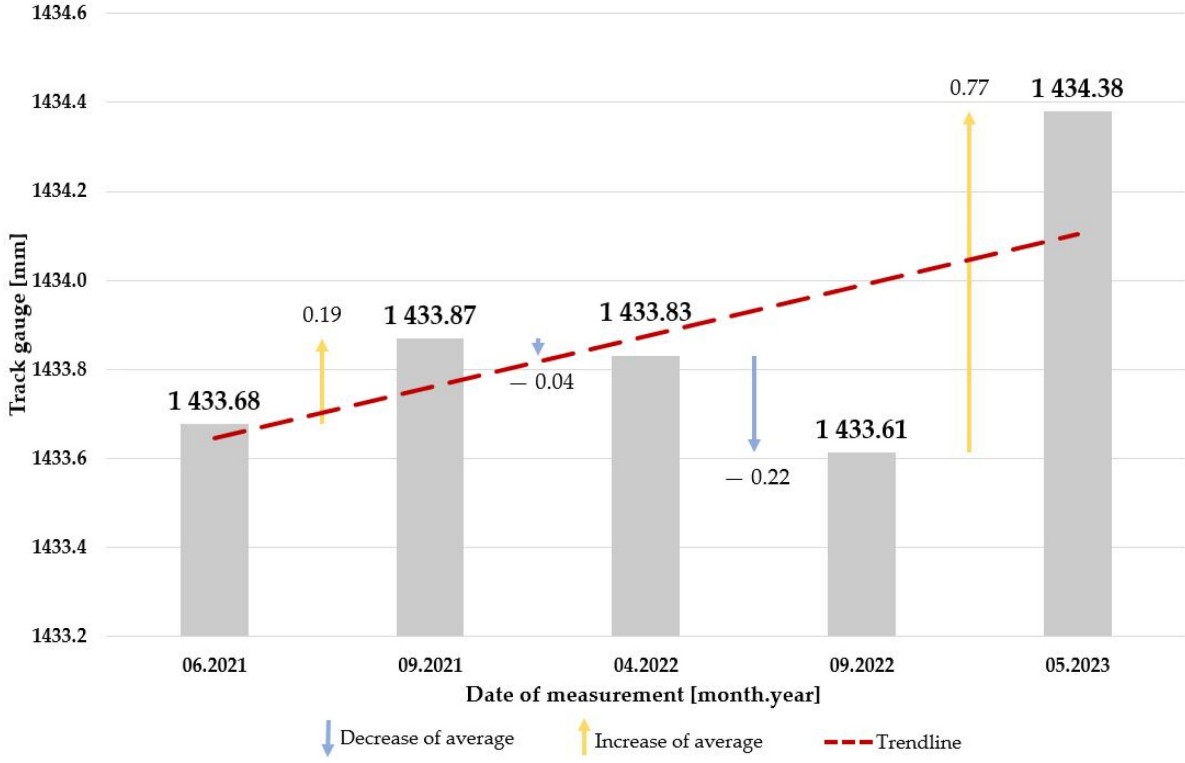

**Figure A14.** The change in the average values for track gauge in Section #3 (ESCRB I).

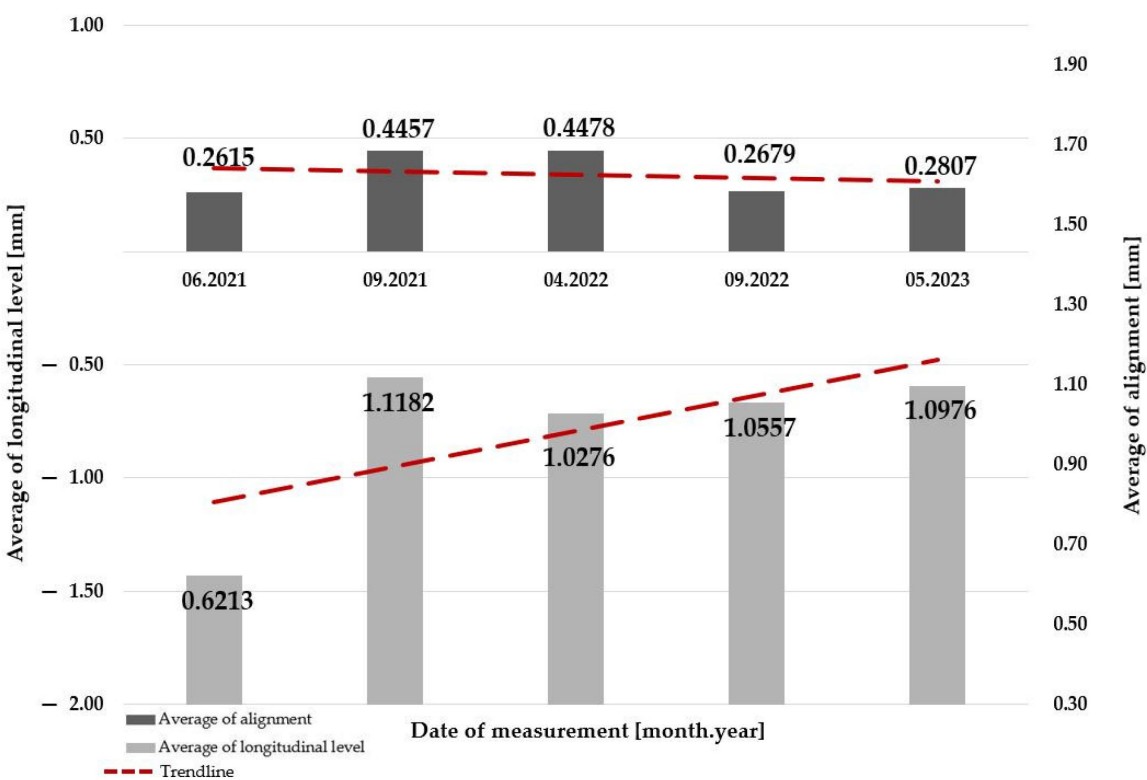

**Figure A15.** The change in the average values for longitudinal level and alignment in Section #3 (ESCRB I).

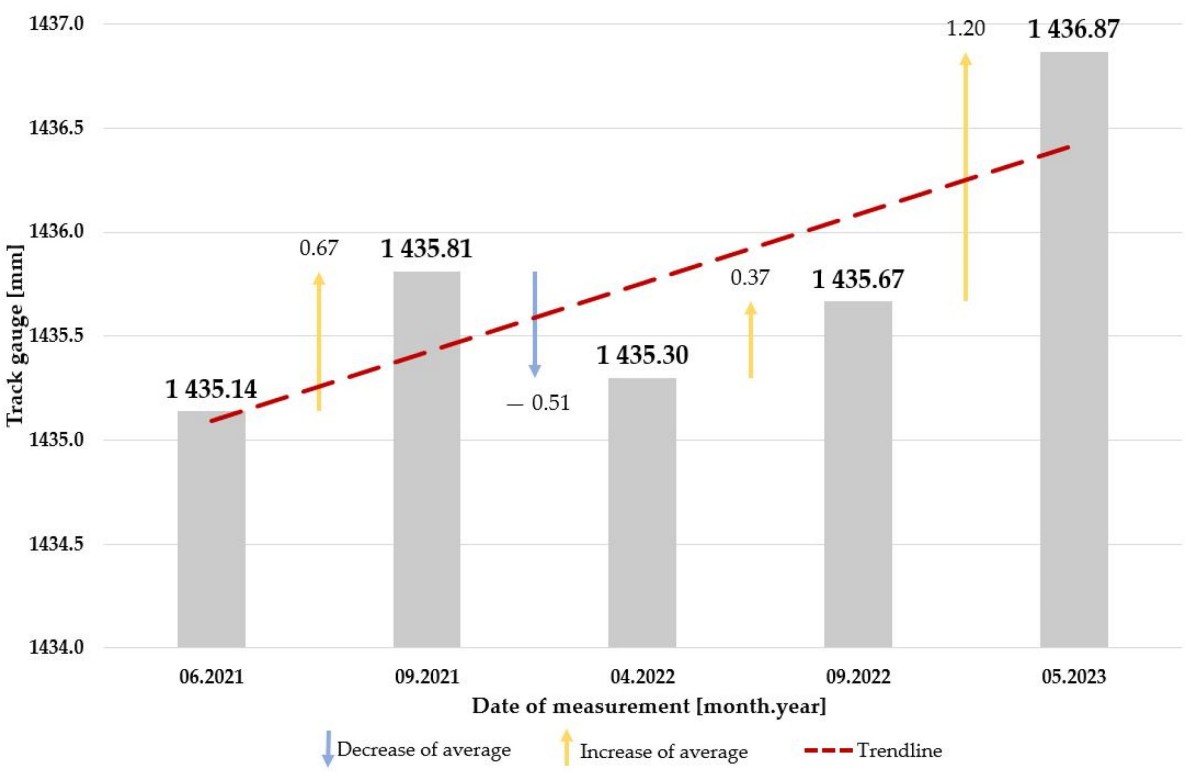

**Figure A16.** The change in the average values for track gauge in Section #4 (ESCRB II).

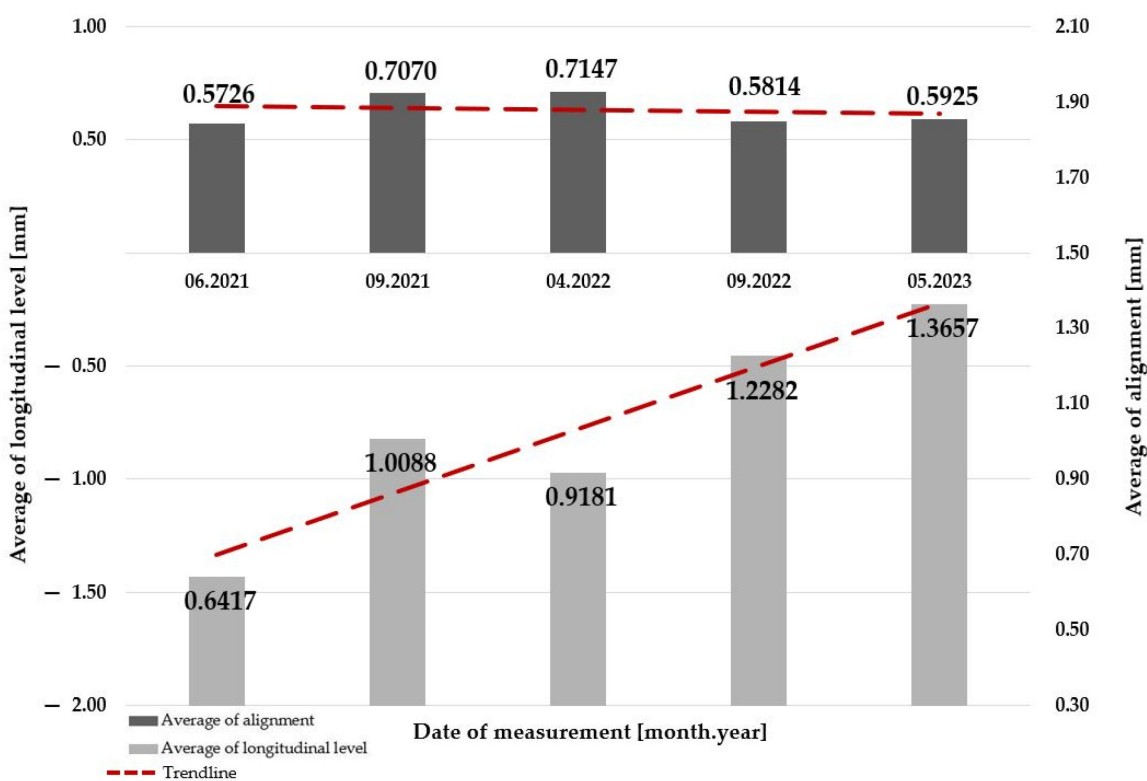

**Figure A17.** The change in the average values for longitudinal level and alignment in Section #4 (ESCRB II).

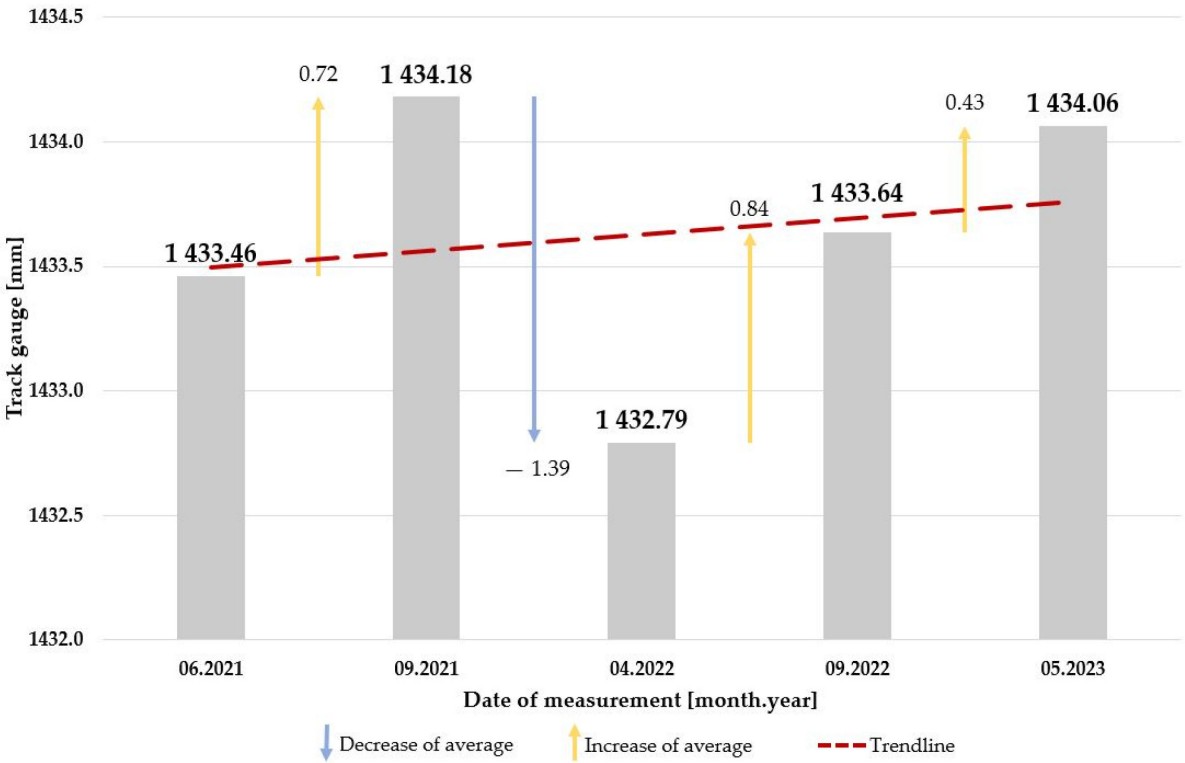

**Figure A18.** The change in the average values for track gauge in Section #5 (ESCRB II).

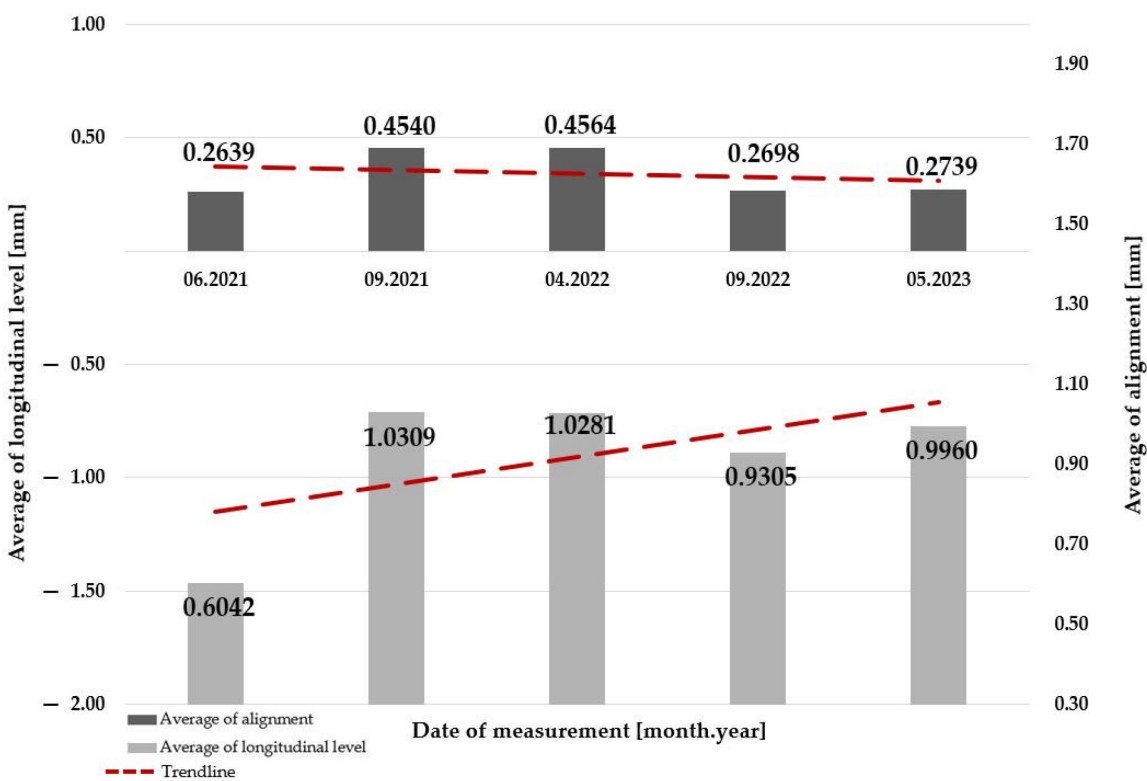

**Figure A19.** The change in the average values for longitudinal level and alignment in Section #5 (ESCRB II).

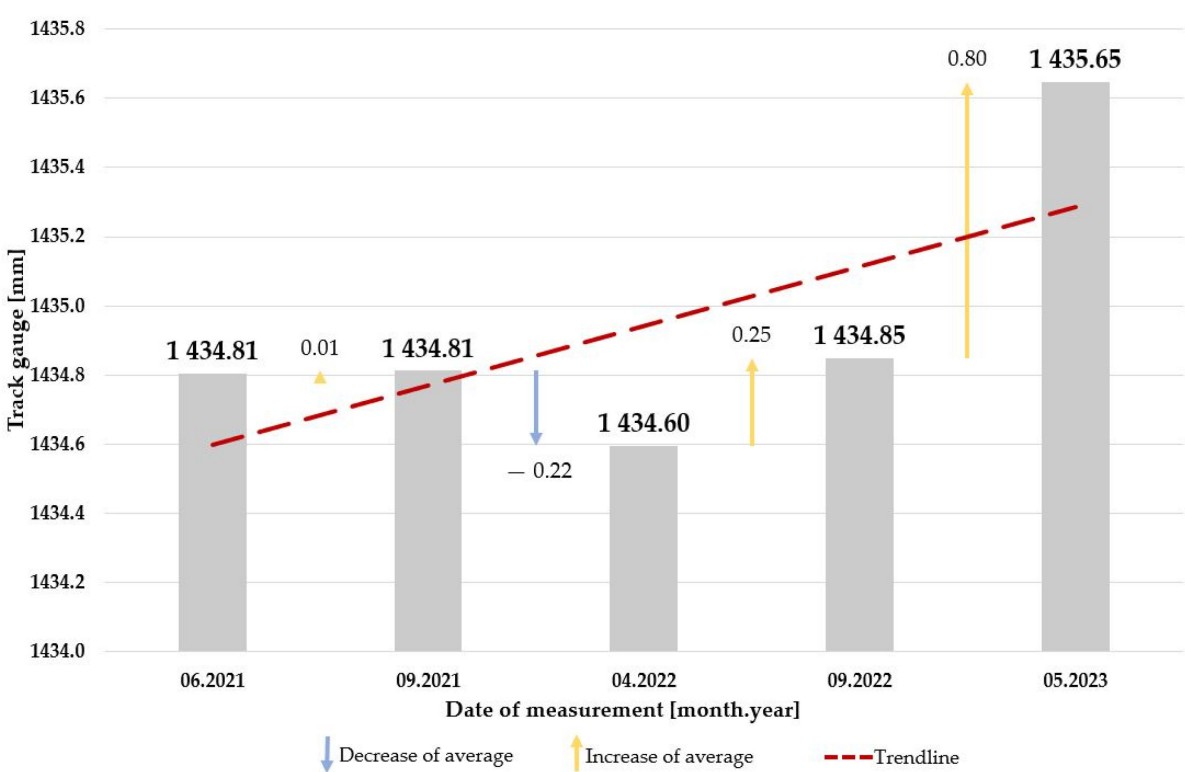

**Figure A20.** The change in the average values for track gauge in Section #6 (ESCRB II).

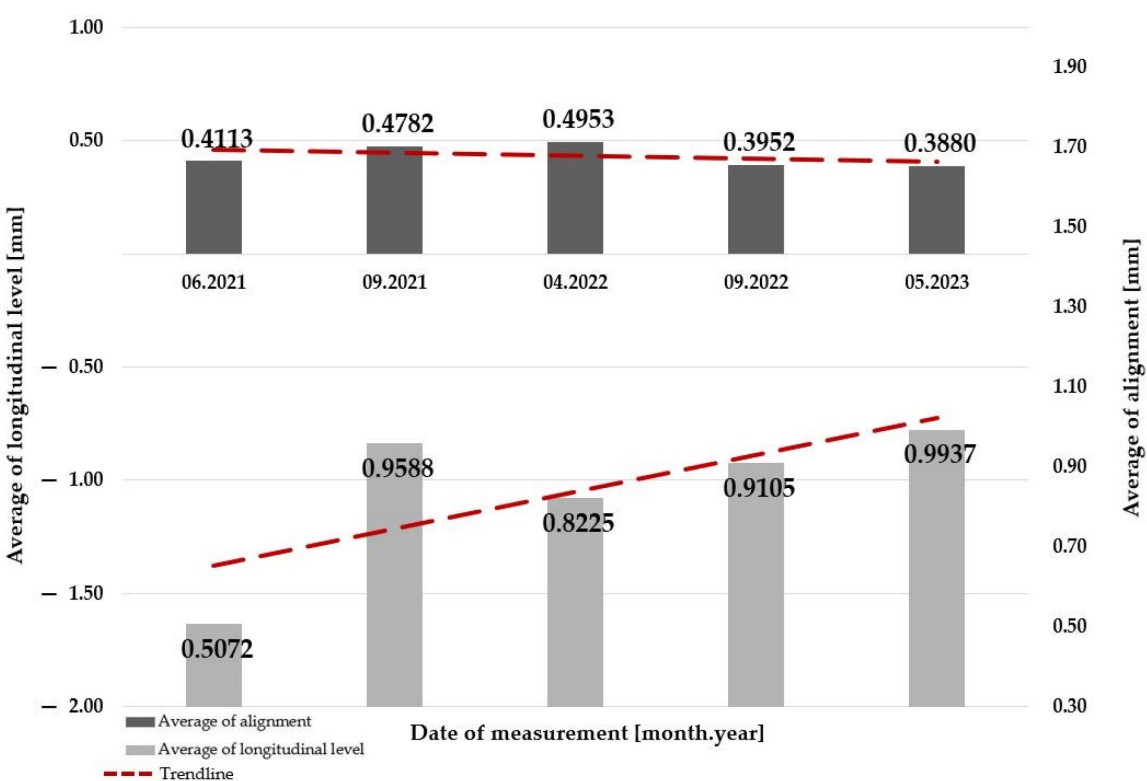

**Figure A21.** The change in the average values for longitudinal level and alignment in Section #6 (ESCRB II).

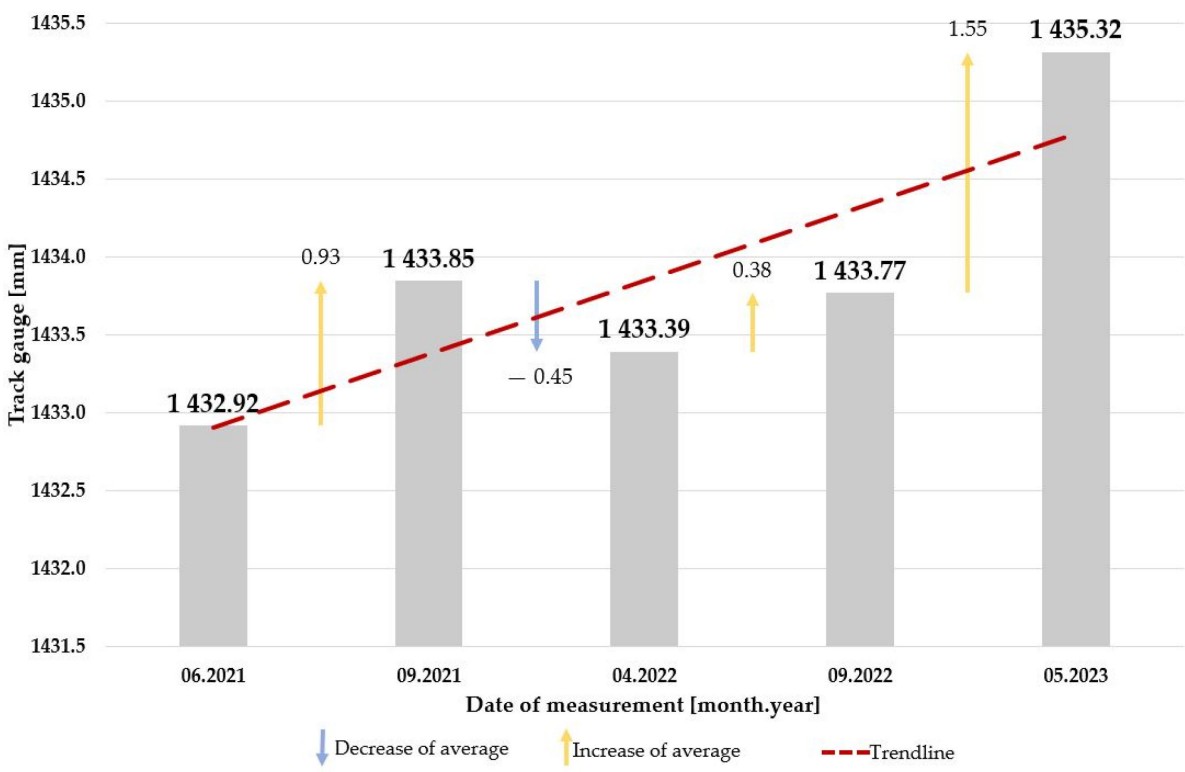

**Figure A22.** The change in the average values for track gauge in Section #7 (ESCRB II).

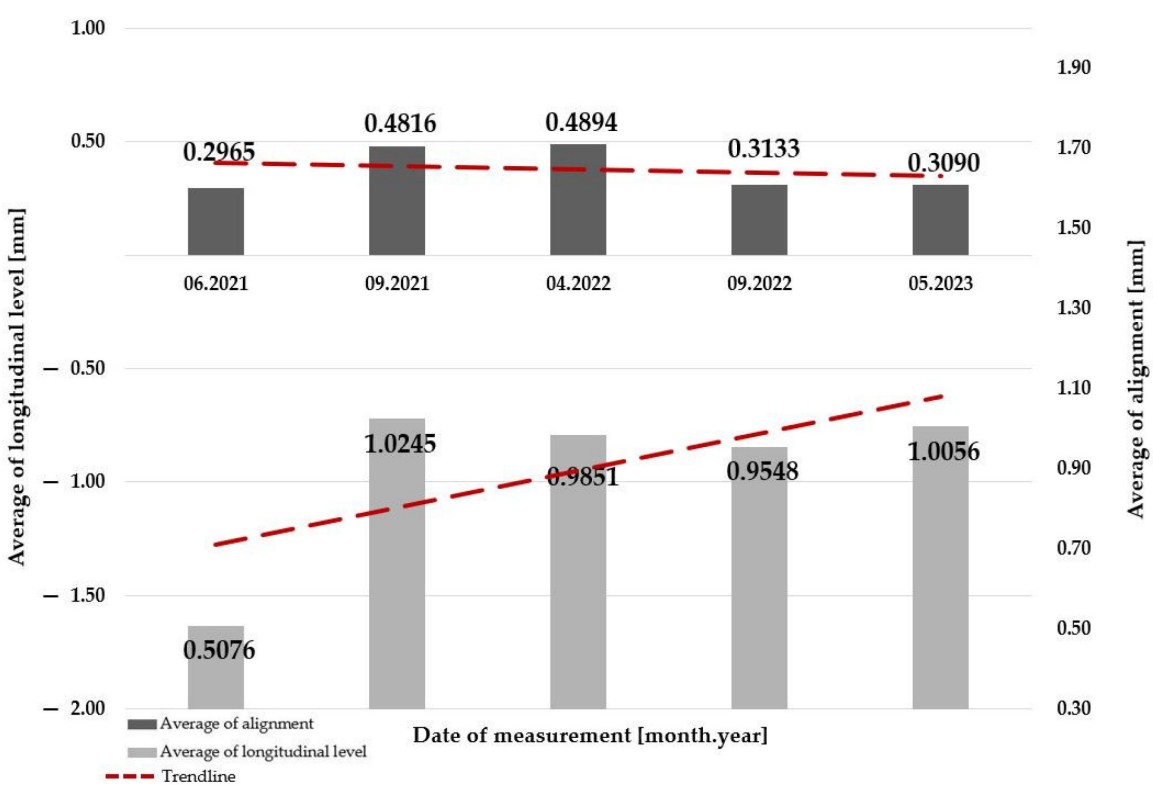

**Figure A23.** The change in the average values for longitudinal level and alignment in Section #7 (ESCRB II).

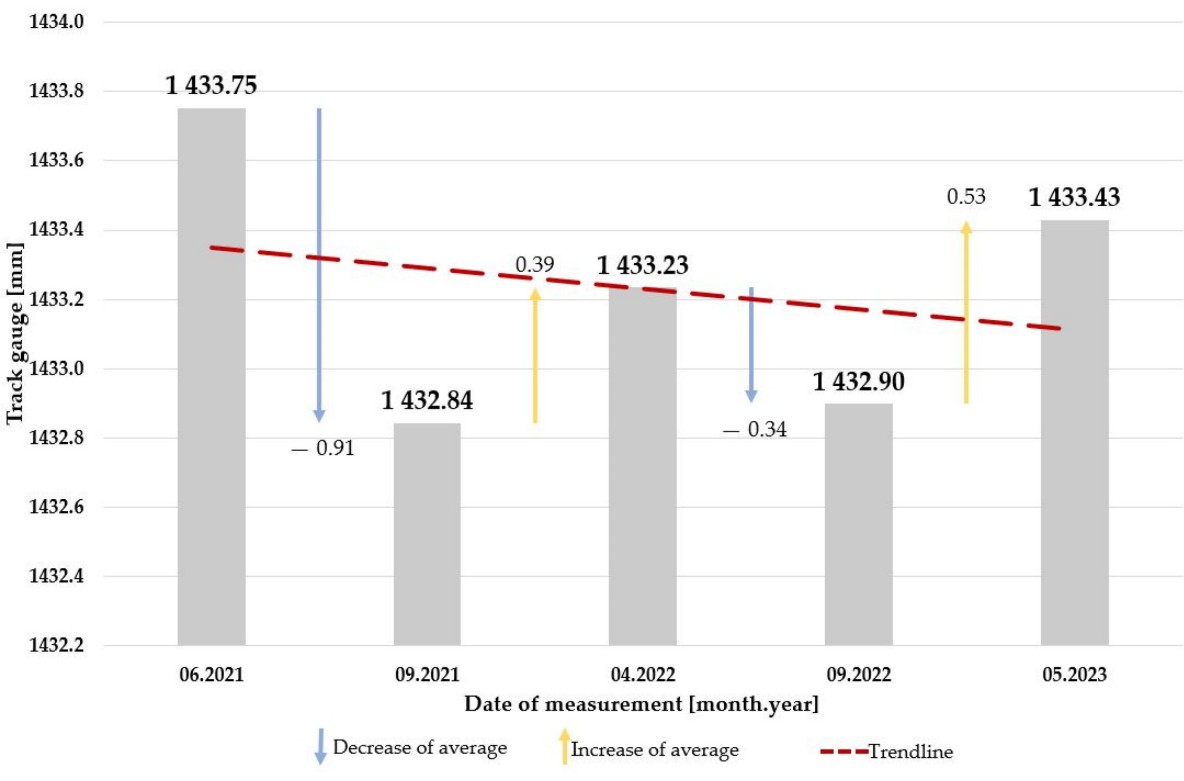

**Figure A24.** The change in the average values for track gauge in Section #8 (ESCRB III).

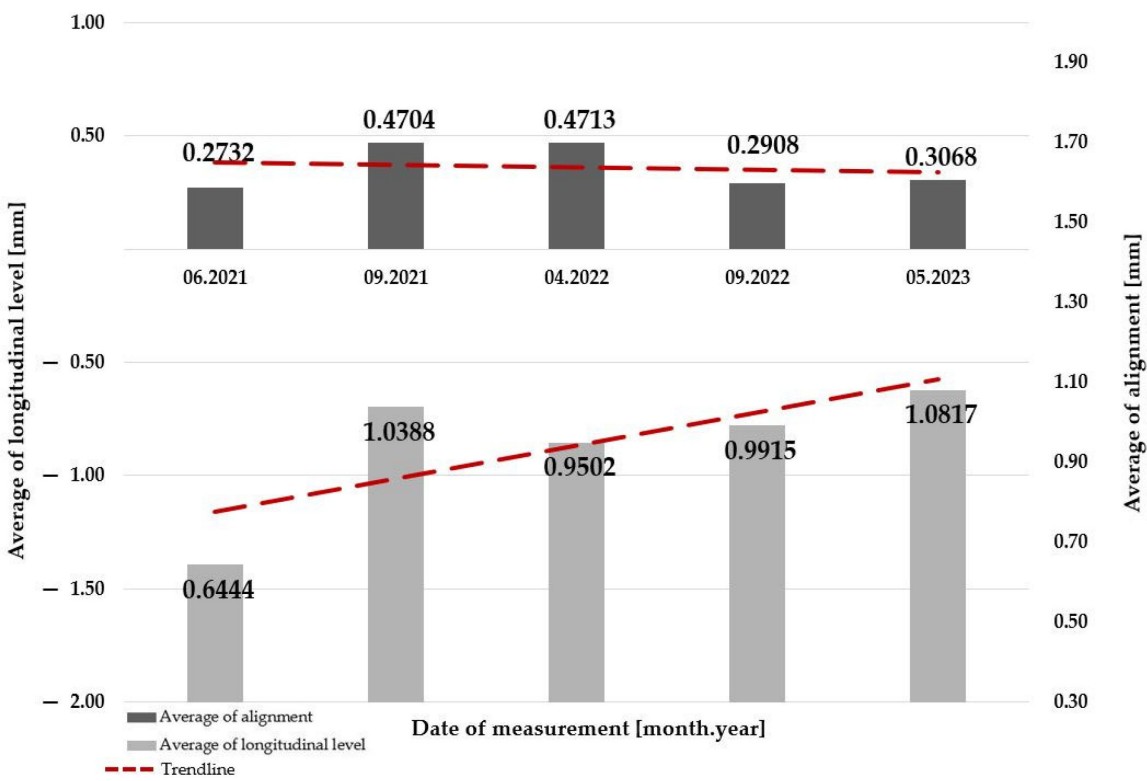

**Figure A25.** The change in the average values for longitudinal level and alignment in Section #8 (ESCRB III).

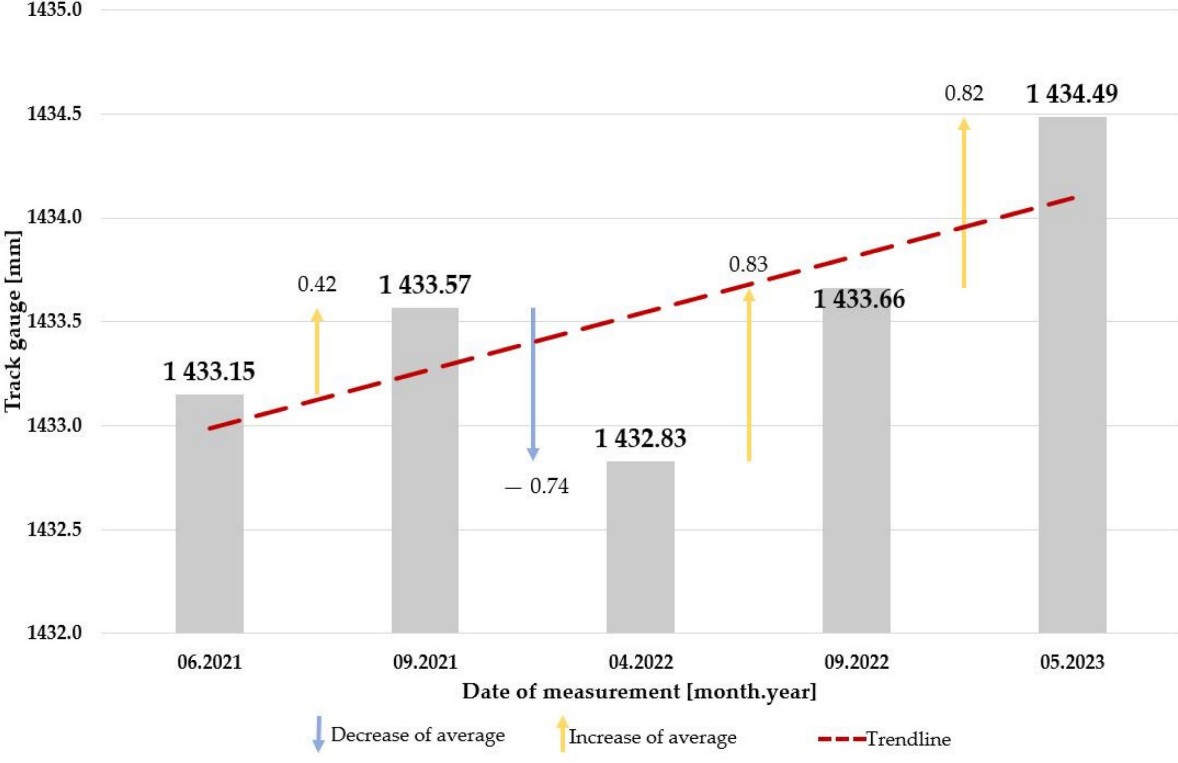

**Figure A26.** The change in the average values for track gauge in Section #9 (ESCRB III).

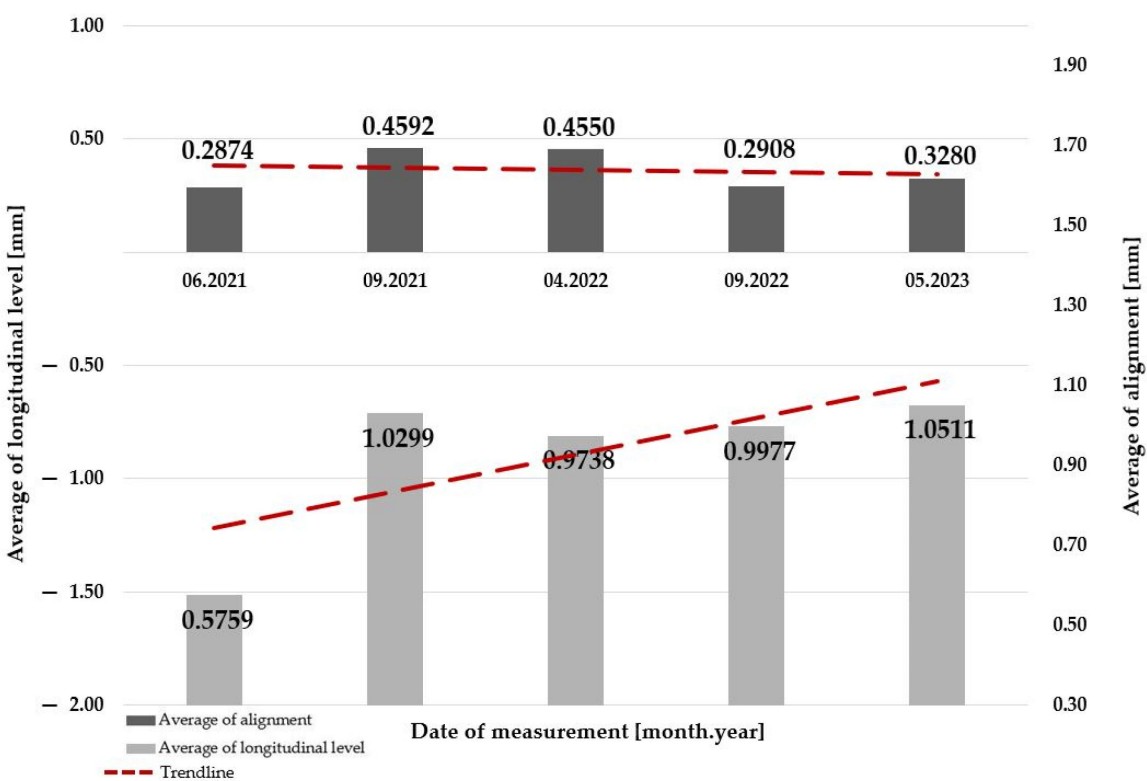

**Figure A27.** The change in the average values for longitudinal level and alignment in Section #9 (ESCRB III).

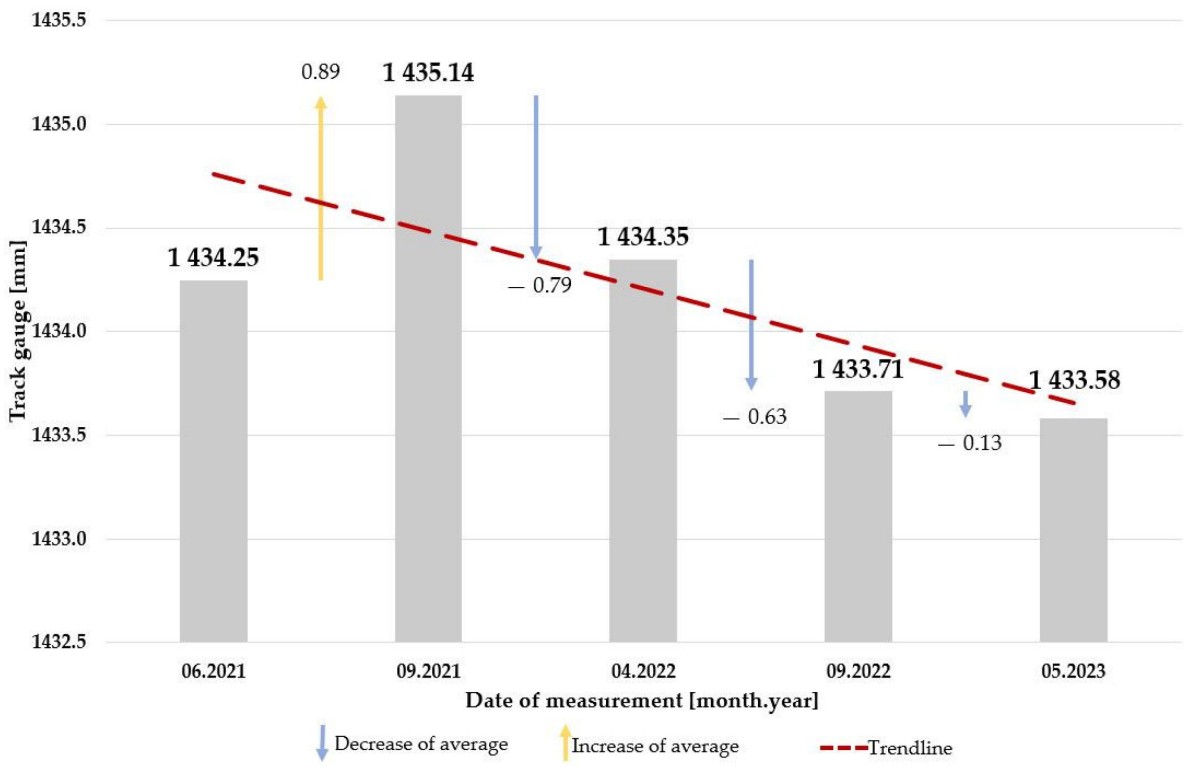

**Figure A28.** The change in the average values for track gauge in Section #10 (ESCRB III).

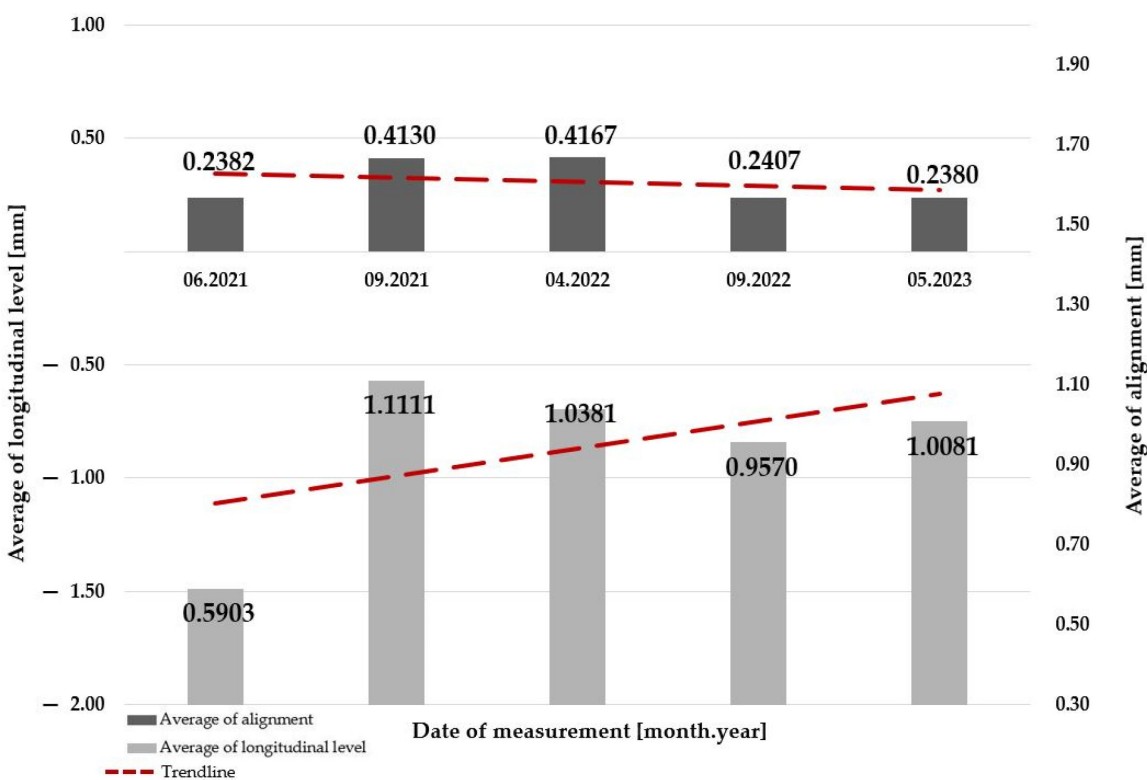

**Figure A29.** The change in the average values for longitudinal level and alignment in Section #10 (ESCRB III).

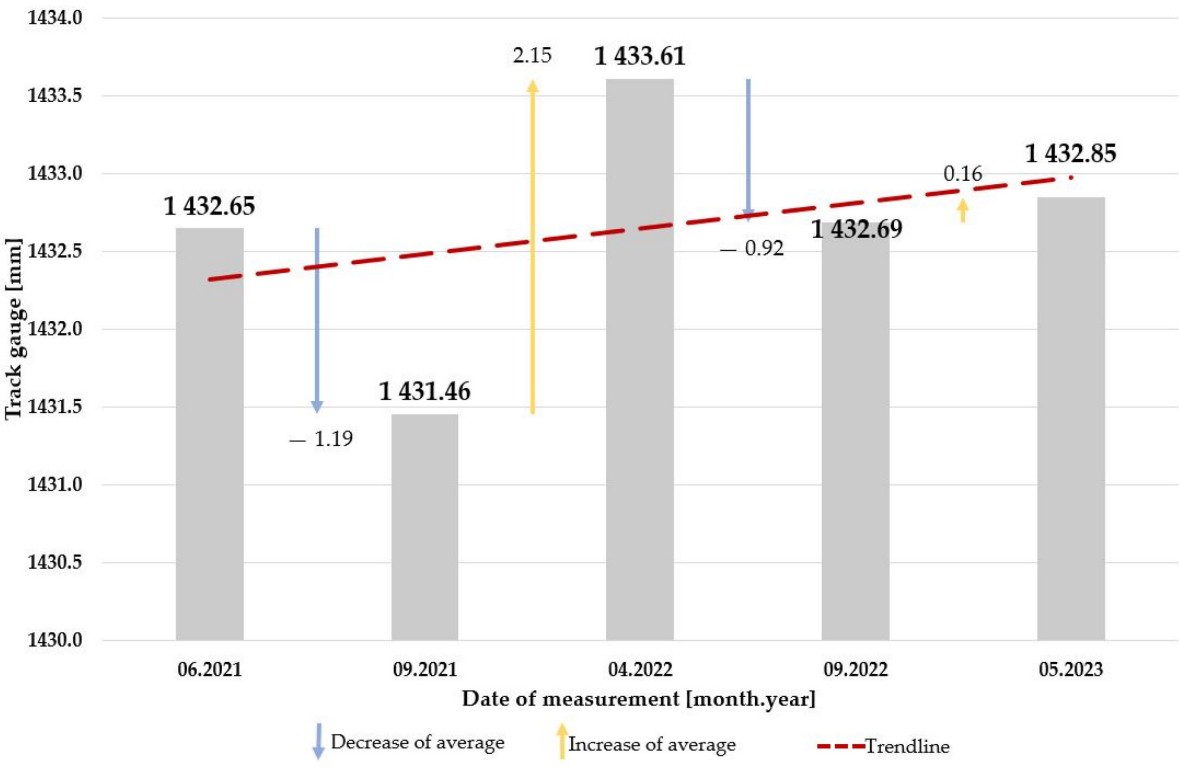

**Figure A30.** The change in the average values for track gauge in Section #11 (ESCRB III).

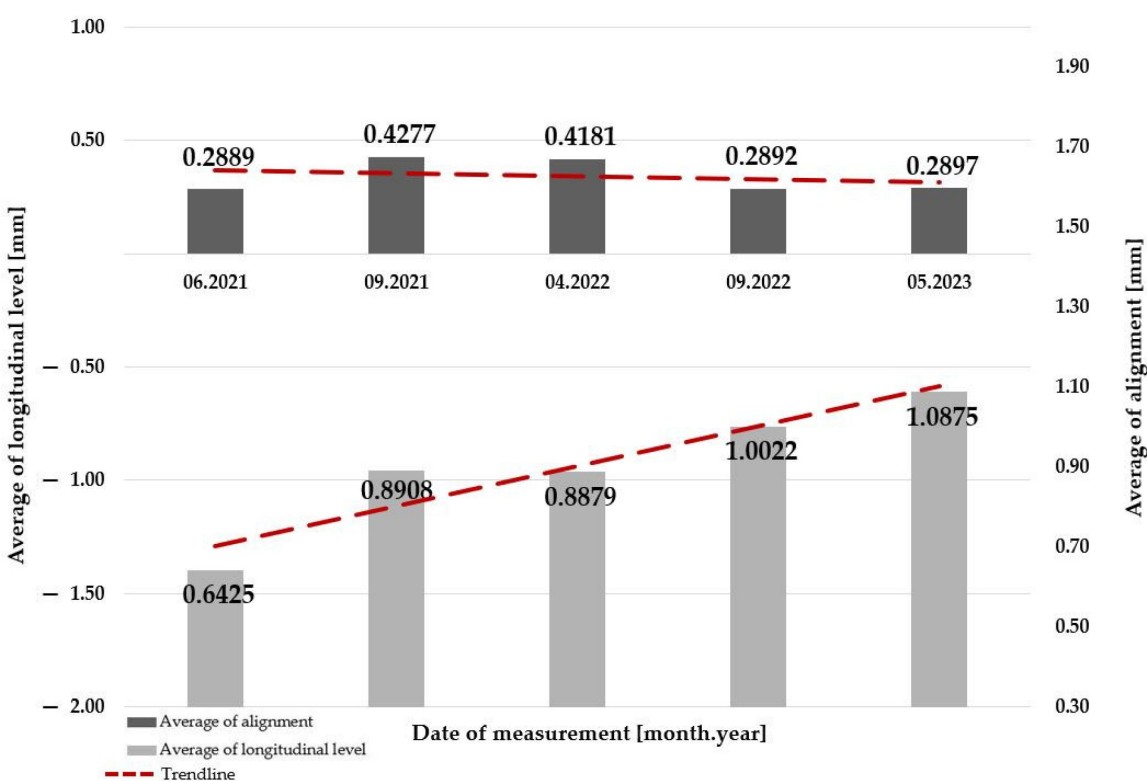

**Figure A31.** The change in the average values for longitudinal level and alignment in Section #11 (ESCRB III).

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
