# Peer review of "Investigation of the Geometrical Deterioration of Paved Superstructure Tramway Tracks in Budapest (Hungary)"

_infrastructures, doi:10.3390/infrastructures8080126_

Round 1

Reviewer 1 Report

Dear Authors,

Thank you for conducting a very interesting study to publish it in an international journal. In my opinion the publication is too broad - please consider this. You may want to make it publish in two parts - with the Editor's approval.

Thanks

Further information on various issues identified in the manuscript appears below:

1.               In my opinion the publication is too broad - please consider this. Perhaps:

(a) it is worth making it publish in two parts - with the Editor's approval.

(b) move selected illustrations or research results to appendices - then there would be an appropriate volume balance between chapters.

2.        I propose to slim down the title of the publication. Do not show the name of the city - Budapest. Already the name of the country is enough, and perhaps not to show it. This is what the abstract, introduction and characteristics of the object of research are for. The current title is restricted by name to Budapest and Hungary. And the content is very interesting - for the global market. I propose to modify it - to make it more optimal and promotional in its entire content. I propose to consider.

3.           Stylistic and punctuation improvements should be made.

4.           A table should not be included in the conclusions. Editorial changes are recommended.

This completes the review.
Kind regards

Author Response

See the attached PDF file in which you can check all the changes tracked.

Reviewer 2 Report

The paper is extremely reach of information and potentially very useful for future studies in this field. Moinor suggestions for further improvementa are the following:

1. Introduction. A reduction would incre ase its readabilityby: i) better focusing the general introduction to tramways eliminating general and historical concepts on transport in general; ii) reducing the references by eliminating those that are dealing with general tramways operation and planning

2. Material and Methods. It is very fine, no specific suggestions. Just to add a map of Budapest railways network could be helpful

3. Results. An effort of synthesis is necessary, in preparation of the following Discussion (section 4). At least a global table and global map of results in a single diagram would be helpful.

4. Discussion. It could be useful a cross analysis among the results of the analysed sections by highlighting differences and similarities among them and deriving possible motivations and remeding actions for degradations.

5. Conclusions. You correctly affirmed that general conclusions are not yet possible but you could just sketch some potential remeding actions to mitigate the effects, also by cross analysis with results of the vaste analysed bibliography.

Author Response

(The authors gave the same response as above.)

Reviewer 3 Report

1. the article name: Examination of the deterioration of “paved” superstructure 2 tramway tracks in Budapest,  should be modified, for example, deterioation just for 3 years of slab track is not ok, and paved, fancy words make trouble to readers.

2. the content should be shorten in pages, the work and findings are not that long.

3. too many self-citations.

2. the content should be shorten in pages, the work and findings are not that long.

please follow style of IMMRD journal paper stlye

Author Response

(The authors gave the same response as above.)

Round 2

Reviewer 3 Report

In this study, the authors examine the geometric evolution of many ESCRB super-structure systems. The TrackScan 4.01 device was used to conduct the measurements in one year. Examined track parameters consist of track gauge, alignment, and longitudinal level, which are measured and recorded parameters.

there are following things should be well handled,

1. there is just one new structure for the application, and the after-operation measuring is just one year, and the track geometry is not highly required compared to HSR or Heavy Haul Railway system, but the paper with 40 pages and almost 100 reference paper, and it against normal stlyle.

2. TrackScan 4.01 device innovation and importance in the paper, for this kind of trolly?

Author Response

See the attached PDF file, in which you can find a comparison part with the previously submitted version.

Round 3

Reviewer 3 Report

too much self cited papers and are not related with the article topic,  the name of the article is not clear to the topic, and the history results of tram is not too long enough.

Author Response

Our answers can be found in the attached PDF document. Please check and consider it. We trust you will be satisfied with the improvements.
